# Transformer as a hippocampal memory consolidation model based on NMDAR-inspired nonlinearity

**Dong-Kyum Kim**[1*]   **Jea Kwon**[2*]   **Meeyoung Cha**[1,3†]   **C. Justin Lee**[2†]
[1]IBS Data Science Group
[2]IBS Center for Cognition and Sociality
[3]KAIST School of Computing
{kdkyum,jeakwon,mcha,cjl}@ibs.re.kr

## Abstract

The hippocampus plays a critical role in learning, memory, and spatial representation, processes that depend on the NMDA receptor (NMDAR). Inspired by recent findings that compare deep learning models to the hippocampus, we propose a new nonlinear activation function that mimics NMDAR dynamics. NMDAR-like nonlinearity shifts short-term working memory into long-term reference memory in transformers, thus enhancing a process that is similar to memory consolidation in the mammalian brain. We design a navigation task assessing these two memory functions and show that manipulating the activation function (i.e., mimicking the $Mg^{2+}$-gating of NMDAR) disrupts long-term memory processes. Our experiments suggest that place cell-like functions and reference memory reside in the feed-forward network layer of transformers and that nonlinearity drives these processes. We discuss the role of NMDAR-like nonlinearity in establishing this striking resemblance between transformer architecture and hippocampal spatial representation.

## 1 Introduction

Synaptic plasticity in the hippocampus is crucial for converting short-term memories into long-term memories during memory consolidation [1–3]. The N-methyl-D-aspartic acid receptor (NMDAR) is essential for mediating this memory formation as well as spatial representation. NMDAR serves as a switch for such plasticity and long-term memory formation [4–6]. Hippocampal place cells [7] and entorhinal cortex grid cells [8] are known to be central for spatial navigation in animals. NMDAR has been highlighted for its importance in place cell representations through hippocampal CA1 neurons [9, 3]. These discoveries have led to a deeper understanding of hippocampal function, inspiring recent efforts to replicate such spatial representation mechanisms in deep neural networks [10–12]. However, whether non-linear dynamics resembling hippocampal functions can be developed and used to support spatial representation in deep learning models remains unclear.

NMDAR is a post-synaptic ion channel that is characterized by nonlinear dynamics that distinguish it from other ion channels in the brain. These nonlinear dynamics are evident in the whole-cell current voltage (I-V) relationship (Fig. 1a) and are modulated by $Mg^{2+}$ ion blockade at the pore region of the channel. Previous research indicates that the $Mg^{2+}$-dependent nonlinear dynamics of NMDAR play a key role in synaptic plasticity and memory formation [5, 6].

Recently, transformer-based deep learning models have been reported to have functions that resemble hippocampal formations [13]. Transformers comprise two consecutive modules: a self-attention

---

[*]Equal contribution.
[†]Corresponding authors.

37th Conference on Neural Information Processing Systems (NeurIPS 2023).

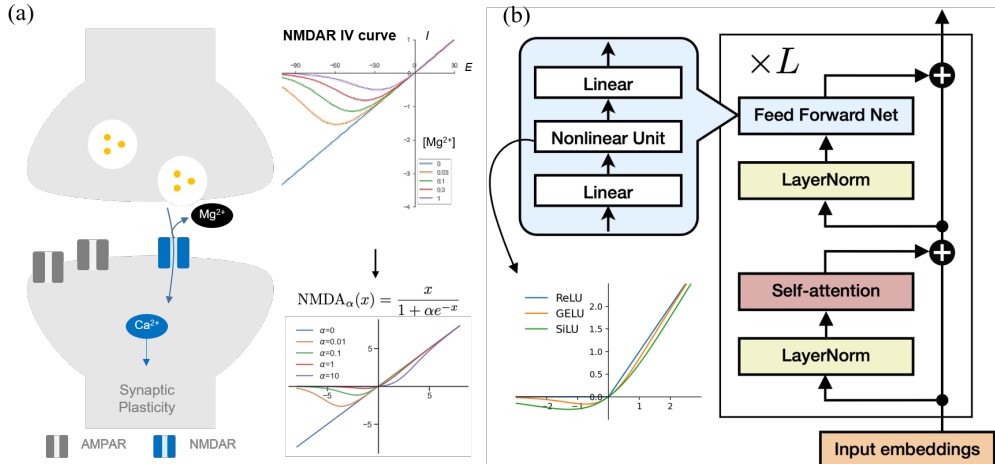

Figure 1: (a) Schematic diagram of $Mg^{2+}$-gated NMDAR modulating synaptic plasticity (left), current-voltage dynamics (I-V curve; top right) and an NMDAR-inspired activation function, $NMDA_\alpha(x)$ (bottom right). (b) Transformer architecture and nonlinear activation functions in its feed-forward network (bottom left): ReLU, Gaussian Error Linear Unit (GELU), and Sigmoid Linear Unit (SiLU).

layer and a feed-forward network (FFN; see Fig. 1b). The self-attention layer is closely related to a recent neuroscience model that bridges with transformer [14] and it has been argued that softmax neurons in this layer behave like place cells in a navigation task [12]. However, the role of neurons in FFNs involved in a spatial navigation task has yet to be elucidated and compared to hippocampal characteristics.

This paper uncovers a resemblance between the NMDAR nonlinearity and recently developed activation functions commonly used in FFNs of deep learning models (Fig. 1). NMDAR functions operate on activity-dependent repulsion of $Mg^{2+}$ ions [15, 16] and this phenomenon is particularly interesting because it supports self-gating of information flow ($Ca^{2+}$ ion influx) in the post-synaptic region. Similar to this NMDAR activity-dependent gating mechanism, activation functions in modern neural networks combine input with a self-gating function (i.e., a sigmoidal function that ranges between 0 to 1) that determines information flow. Our observation prompted the following inquiry: **Can NMDAR-like nonlinearity in the feed-forward network layer of transformers enhance the formation of long-term memory and spatial representation by place cells?**

Here we derive a novel NMDAR-like activation function using the NMDAR current-voltage (I-V) curve. In parallel we design a spatial navigation task in a 2D grid environment that assesses two memory types well-defined by neuroscience research: working memory and reference memory [17, 18]. The former assesses events from within-trial, while the latter controls across trials in a stable (unchanging) environment. We then test the transformer model with the NMDAR-like activation function and find that 1) place cell representations emerge in the FFN layer rather than in the self-attention layer, 2) the nonlinearity of the NMDAR-like activation function can regulate reference memory, 3) place cell-like neurons in FFNs are strongly correlated with reference memory, whereas this correlation is not observed in the self-attention layer; and 4) NMDAR-like activation shows the best reference memory performance when compared to other available activation functions.

Collectively, these findings suggest that adopting NMDAR-like nonlinearity in FFN of transformer models can enhance the formation of long-term memory and spatial place cell representation, similar to previous observations in the animal brain. We believe these findings have important implications for developing new brain-inspired AI models and for understanding similar processes that occur in the brain and AI models.

## 2 Methods

### 2.1 Designing a 2D navigation task to test the role of working memory and reference memory

We designed a sensory observation prediction task in which an agent randomly walks in a 2D grid environment and is trained to predict subsequent sensory observations (see Fig. 2a) [12]. The agent

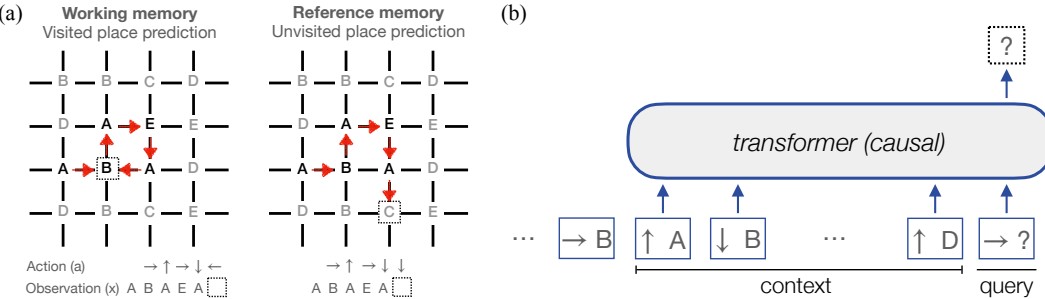

Figure 2: (a) Sensory observation prediction task in a 2D grid, where dotted squares indicate the target position to predict given a sequence of past actions and observations. The unvisited and visited places are represented in gray and black blocks, respectively. (b) A transformer model for predicting the location of an upcoming sensory observation based on sequences of [Action ($a$), Observation ($x$)] pairs. Using a sequence of pairs in the context, the model is trained to predict the masked observation (i.e., the subsequent observation) corresponding to the final query action.

receives a sequence of previous [Action ($a$), Observation ($x$)] pairs as input, and its goal is to predict the next observation, which is masked (i.e., dotted squares in the sequence of Observation ($x$) in the figure). We use the transformer architecture as our model.

We generated $N$ maps of 11×11 2D grids. A random sensory observation chosen from ten letters is placed at each position on each map. Agents can move 'up,' 'right,' 'down,' 'left,' or 'stay.' An agent starts at a random position and initiates a random walk on the map, which is randomly selected from $N$ training maps, for 2,048 steps for each trial.

Our design novelty is the consideration of two memory types: **short-term working memory** and **long-term reference memory**. When the prediction based on nodes that were previously visited during the random walk is incorrect, it counts as a *working memory error* (Fig. 2a left). In contrast, when the prediction based on unvisited nodes is incorrect, it counts as a *reference memory error* (Fig. 2a right). See details of the navigation task and definitions in Appendix A.4 and Fig. S3.

It is important to note that minimizing the reference memory error by memorizing input sequences is infeasible; the possible number of sequence configurations is exponential since the input sequence is randomly generated at each trial. To solve this task, the model needs to: 1) understand the abstract structure of 2D space, 2) infer which map it is on from the input sequence data, and 3) memorize the sensory observations and their positions on that map. Compared to the previous works [14, 12] that focused on working memory error, the current paradigm evaluates two distinct types of memory error and gives a more holistic view of model performance.

## 2.2 Transformer: two separate memory systems

We here review the self-attention mechanism and FFNs in the transformer architecture [13] that hypotheses about two separate memory systems are based on—working memory (formed within-trial) and reference memory (formed across-trials)—and why they are inferred to reside in self-attention layers and FFNs, respectively.

**Self-attention mechanism** Given a sequence $\{\mathbf{x}_1, ..., \mathbf{x}_T\}$ of $d$-dimensional input embeddings, the self-attention layer calculates the interaction term between each embedding element within a context window via the self-attention mechanism. More formally, each input embedding applies two linear layers ($W_k$ and $W_v$) to the embeddings to form the key matrix $K$ and value matrix $V$:

$$K^\top = [\mathbf{k}_{t-c}^\top \ \mathbf{k}_{t-c+1}^\top \ \cdots \ \mathbf{k}_t^\top], \quad V^\top = [\mathbf{v}_{t-c}^\top \ \mathbf{v}_{t-c+1}^\top \ \cdots \ \mathbf{v}_t^\top], \tag{1}$$

where $\mathbf{v}_i = \mathbf{x}_i W_v$ ($W_v \in \mathbb{R}^{d \times d_k}$) and $\mathbf{k}_i = \mathbf{x}_i W_k$ ($W_k \in \mathbb{R}^{d \times d_k}$). Here, $c$ denotes the context length. The key matrix $K \in \mathbb{R}^{(c+1) \times d_k}$ is then used to compute the interaction score between an input embedding at step $t$ and all the vectors in $K$ via dot products:

$$\mathbf{s}_t = \mathbf{q}_t K^\top, \quad \text{where } \mathbf{q}_t = \mathbf{x}_t W_q \ (W_q \in \mathbb{R}^{d \times d_k}). \tag{2}$$

The normalized values of $\mathbf{s}_t \in \mathbb{R}^{(c+1)}$, called attention values, are calculated via the softmax function; the final output of the self-attention mechanism is a weighted sum of the value vectors in

Table 1: Comparison of common activation functions with $\text{NMDA}_{\alpha,\beta}$

| $\text{NMDA}_{\alpha,\beta}$ | Name | Equation | Reference |
|---|---|---|---|
| $\text{NMDA}_{\alpha=1,\beta=1}(x)$ | SiLU(x) | $x\sigma(x)$ | [20] |
| $\text{NMDA}_{\alpha=1,\beta=1.702}(x)$ | GELU(x) | $x\sigma(1.702x)$ | [21] |
| $\text{NMDA}_{\alpha=1,\beta=\infty}(x)$ | ReLU(x) | $\max(0,x)$ | [22] |
| $\text{NMDA}_{\alpha=1,\beta}(x)$ | Swish(x) | $x\sigma(\beta x)$ | [23] |

$V \in \mathbb{R}^{(c+1)\times d_k}$ with the attention values:

$$\mathbf{y}_t = \texttt{softmax}\left(\frac{\mathbf{q}_t K^\top}{\sqrt{d_k}}\right)V. \tag{3}$$

After this update, $\mathbf{y}_t \in \mathbb{R}^{d_k}$ is updated by another linear transformation $W_o \in \mathbb{R}^{d_k \times d}$: $\mathbf{z}_t = \mathbf{y}_t W_o$. The output $\mathbf{z}_t$ is added to the $\mathbf{x}_t$; $\mathbf{z}_t + \mathbf{x}_t$ providing the final output of the self-attention layer, and this information is sent through to the subsequent layer.

**Feed-forward networks (FFNs)**   This component consists of two linear layers with a point-wise nonlinear activation function $\phi$:

$$\texttt{FFN}(\mathbf{x}_t) = \phi(\mathbf{x}_t U_1^\top)U_2, \tag{4}$$

where $U_1 \in \mathbb{R}^{d_f \times d}$ and $U_2 \in \mathbb{R}^{d_f \times d}$ are trainable weight matrices. Sukhbaatar et al. [19] showed that Eq. (3) and Eq. (4) have similar structures except for the following: 1) $U_1$ and $U_2$ matrices are fixed over different input sequences while $K$ and $V$ matrices are dynamically changed with input and 2) operations in FFNs are point-wise or local while the self-attention layer has non-local operations, e.g., the softmax function and dot products between different elements. This observation suggests that the FFNs store "general" knowledge about the task that does not depend on the situation.

## 2.3   Resemblance of NMDA receptor nonlinear dynamics with modern activation functions

NMDAR's nonlinear dynamics mostly arise from the voltage-gated $Mg^{2+}$ repulsion at the NMDAR channel's pore [15, 16] (Fig. 1a left). Previous work showed this nonlinear $I$-$V$ relationship to be:

$$I_{\text{norm}} = V\mathbf{p}_{\alpha,\beta}(V) \tag{5}$$

where $V$ represents an input voltage and $\mathbf{p}_{\alpha,\beta}(V)$ is a voltage-dependent channel open probability that follows the ion blockade model [24]:

$$\mathbf{p}_{\alpha,\beta}(V) = \frac{1}{1 + \alpha e^{-\beta V}} \tag{6}$$

where $\alpha = [Mg^{2+}]/K_{Mg^{2+}}$ is a parameter determined by $[Mg^{2+}]$, $K_{Mg^{2+}}$ is a dissociation constant, and $\beta$ is a temperature constant. For further details, see Appendix A.2.

The dynamics of the NMDA receptor closely resemble those of new activation functions such as ReLU or GELU (see Fig. S2 in Appendix A.3). This visual resemblance motivated us to define a new NMDAR-inspired activation function (see details in Appendix A.3) as follows:

$$\text{NMDA}_{\alpha,\beta}(x) = x\mathbf{p}_{\alpha,\beta}(x) = \frac{x}{1 + \alpha e^{-\beta x}}. \tag{7}$$

This $\text{NMDA}_{\alpha,\beta}(x)$ incorporates modern activation functions with varying values of temperature constant, $\beta$ (Table 1). To investigate the $Mg^{2+}$-gated nonlinear dynamics, $\alpha$, we compared various activation functions with $\text{NMDA}_\alpha(x) = x\mathbf{p}_{\alpha,\beta=1}(x)$.

## 2.4   Testing the NMDAR-inspired activation in navigation tasks

The transformer model is trained using softmax cross-entropy loss to predict subsequent sensory observations (i.e., dotted squares in Fig. 2). Instead of using sinusoidal positional encoding [13], we employ recurrent positional embedding which encodes the location of an input element by using the recurrent neural network (RNN); this method is closely related to the most advanced neuroscience model of the hippocampus [12].

We generate the embedding vectors of the sensory observation sequence with a word embedding layer, but the embedding vectors of the action sequence are generated using an RNN; $\mathbf{e}_{t+1} = \tanh(\mathbf{e}_t W_a)$, where $\mathbf{e}_t$ is a recurrent positional embedding at step $t$, and $W_a$ is the action-dependent trainable weight matrix. The input is given by $\{[\mathbf{e}_1, \mathbf{x}_1], [\mathbf{e}_2, \mathbf{x}_2], \ldots, [\mathbf{e}_t, \mathbf{x}_t]\}$, where $\mathbf{x}$ denotes the embedding vector of sensory observation $x$; the initial recurrent positional embedding $\mathbf{e}_1$ is sampled from a normal distribution, and we mask the last observation $x_t$.

In our experiment, FFN in the transformer model consists of two linear layers (see Fig. 1b and Eq. (4)) with the NMDAR-inspired activation function $\text{NMDA}_\alpha$. We use TransformerXL [25] with an extended memory length of 32 and segment length of 32 so that the context length $c$ is 64 and working memory error is measured when the node to predict its sensory observation is in the context window (see Fig. 2b); i.e., a node that the agent has not been visited in the last 64 steps is treated as *an unvisited node*. Note that our model is unable to access the sensory observations of unvisited nodes via the self-attention mechanism due to the fixed context window size.

The input embedding is a concatenated vector $[\mathbf{e}, \mathbf{x}]$ of the word embedding $\mathbf{x}$ (dimension of 256), and the recurrent positional embedding $\mathbf{e}$ (dimension of 256) so that the total input embedding dimension is 512. The number of heads in the self-attention layer is 8, and the number of neurons in the FFN is 2,048. The dropout rate is set to 0.1, and the maximum clip norm of the gradient is set to 0.25. We employ the ADAM optimizer [26] and a learning rate schedule with a linear decay from 0.0001 (start) to 0 (end). We run 512 random walk simulations (trials) in parallel to collect training trajectories. The total number of random walking steps is 2,048 for each simulation, and the total number of steps for training a model is 512 (batch size; the number of trials per epoch) $\times$ 2,048 (total number of steps in a trial) $\times$ 200 (number of epochs) (see Fig. S3 in Appendix A.4). The average number of unvisited nodes in a single trial was 561. The PyTorch code for reproducing our results is available at `https://github.com/kdkyum/transformer_memory_consolidation`.

## 3 Results

### 3.1 Working memory and reference memory errors

To measure the impact of nonlinearity $\alpha$ in the FNNs, we trained transformer models with different values of $\alpha$ in $[0, 0.01, 0.05, 0.1, 0.5, 1, 5, 10]$ and evaluated the working memory and reference memory errors on the train maps (i.e., familiar maps) and test maps (i.e., novel maps). The changing $\alpha$ values in a transformer model mimic the changes in NMDAR $\text{Mg}^{2+}$-gating in the brain, as inspired by neuroscience findings that selective inhibition of hippocampal CA1 NMDAR can disrupt the consolidation of short-term working memory to long-term reference memory.

The top left plot in Fig. 3a shows that the reference memory error on the training maps rapidly decreased over training trials when $\alpha$ is larger than zero, with a larger improvement for increasing $\alpha$. The reference memory error on the novel maps, however, was nearly constant at a chance level of $0.9 \; (= 1 - 1/(\text{number of letters}))$ for all $\alpha$ (see Fig. 3a top right). Fig. 3a bottom right shows that working memory was active on novel maps that had not previously been shown during training. This finding suggests that working memory formation is intact on novel maps. Training the models on different numbers of maps, $N$, Fig. 3b shows that increasing nonlinearity (i.e., $\alpha$) helped to activate reference memory, and the trend of improvement was consistent, as shown for $N = 32, 48$, and $64$. Training more maps led to larger reference memory errors, because increasing $N$ means the model needs to memorize what-where pairs (i.e., each training contains unique what-where information).

In addition, we find that removing nonlinearity in the NMDAR-inspired activation function ($\alpha = 0$) impairs reference memory formation (Fig. 3b) but leaves working memory formation intact (Fig. 3a). This means that even though the trainable parameters exist in the self-attention layer, a lack of nonlinearity in the feed-forward layer significantly impairs the formation of long-term reference memory in familiar maps. This result suggests that short-term working memory and long-term reference memory are physically stored in separate structures (the self-attention layer and the FFN) of transformers and possibly gated by the nonlinear activation functions residing in the FFN.

We next tested other nonlinear activation functions, including GELU ($x\sigma(1.702x)$), ReLU ($\max(0, x)$), LeakyReLU ($\max(0, x) + 0.01\min(0, x)$), sigmoid, and tanh. Fig. 3c shows that the newly proposed biologically-inspired function ($\text{NMDA}_{\alpha=10}$) yielded substantially lower reference memory errors relative to other activation functions. A comparison with the second-best-performing

GELU activation, for example, indicated a statistically significant difference with $p$-value less than 0.05 (see Appendix A.6). These results open up the possibility of finding a better activation function through a new $\alpha$ hyperparameter space of the NMDA-inspired activation function.

Expanding on the evaluation, one may consider other memory types like recurrent positional embedding instead of reference memory. We used non-recurrent positional embeddings to train the models (see Appendix A.5) and confirmed that working memory and reference memory errors increased more substantially. This finding supports the idea that working memory is crucial for memory consolidation and that disrupting it impairs reference memory. However, we also saw a similar trend of decreasing reference memory error while increasing $\alpha$ of $\text{NMDA}_\alpha$ (see Fig. S4).

We further assessed the prediction error of the first visited node. While the reference memory error is defined as a prediction error on a node that the agent has not visited in the previous 65 steps, the first visited node prediction error is a prediction error on a node that the agent visits for the first time in a trial. The results for the first visited node prediction error were identical to the results for the reference memory error (see Fig. S6 in Appendix A.5). These findings suggest that reference memory is used in training maps to predict the unvisited node.

## 3.2 Place cells in feed-forward networks

One of the most striking characteristics of the hippocampus is the presence of the **place cells** [7], which are neurons in the brain that fire at a particular place in the environment [7]. Studies have shown that hippocampal place cells encode spatial location through localized firing patterns. They have been considered a substrate for long-term memory of the location where specific events occurred (e.g., a previously visited position in a navigation task). Selective impairment of NMDAR in hippocampal CA1 disrupts place cell emergence and memory consolidation of long-term memory [2, 3, 9].

We investigated whether place cells emerge during the formation of long-term reference memory in a transformer model, building upon our previous results demonstrating selective impairment in long-term reference memory with $\text{NMDA}_\alpha$ modulation. We newly designed a metric called the **place cell score** described in Eq. S1 in Appendix A.1 and examined the spatial representation within the transformer architecture. This metric, ranging from 0 (i.e., homogeneous firing) to 1 (i.e., firing is specific to the agent's position in the grid) quantifies the firing specificity of each neuron for spatial

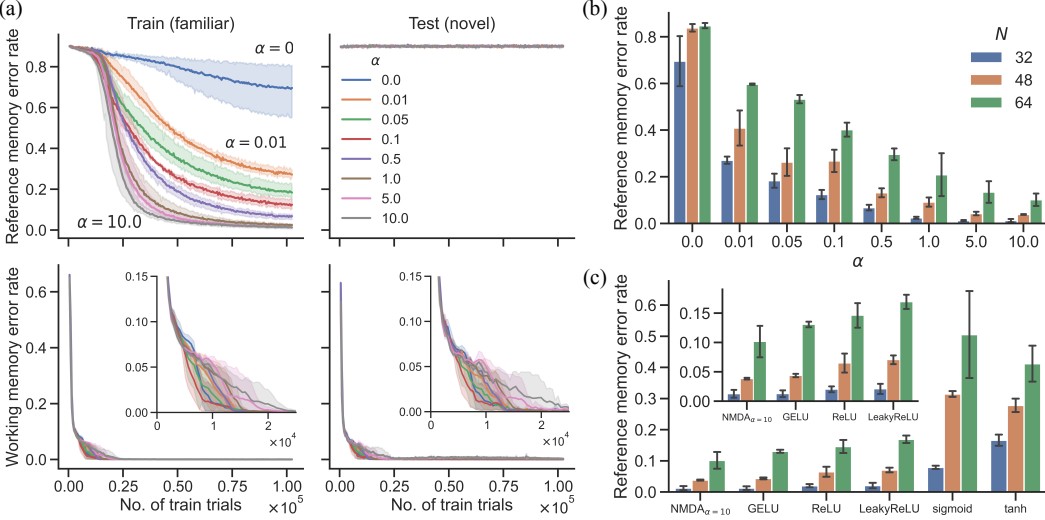

Figure 3: (a) Reference and working memory error rates over training trials for training maps and testing maps for $N = 32$ training maps. Insets in the bottom figures show working memory error rates during the initial training phase. (b) Reference memory errors were evaluated on training maps over different values of $\alpha$ in $\text{NMDA}_\alpha$ and $N$. (c) Reference memory error comparison between $\text{NMDA}_\alpha = 10$, GELU, ReLU, LeakyReLU, sigmoid, and tanh activation functions. Inset: magnified view of the top 4 activation functions. Error bars and shaded areas represent the standard deviation of errors from three independently trained models.

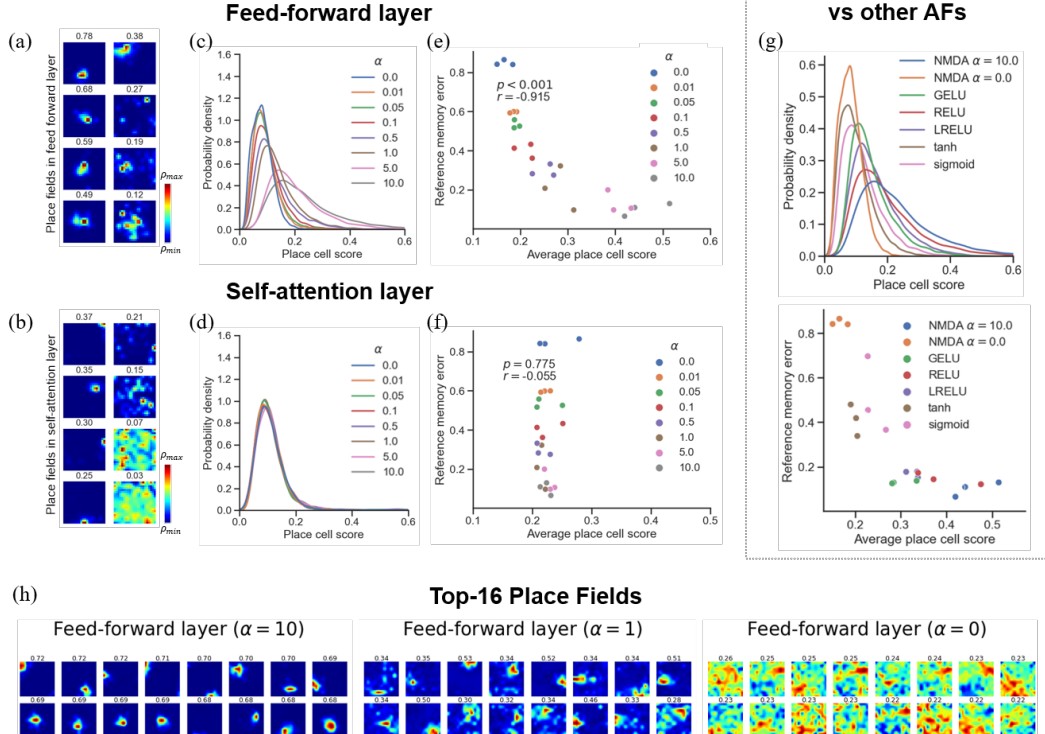

Figure 4: Reference memory-related place cells selectively emerge in FFN but not in the self-attention layer over increasing $\alpha$. (a, b) Example rate maps with place scores in FFN and self-attention layers at $\alpha = 10$; from top left (high) to bottom right (low); color bar indicates the firing rate between $\rho_{\max}$ and $\rho_{\min}$. (c-d) Place cell score distributions with varying $\alpha$ in FFN (c) and self-attention layers (d). (e-f) Scatter plot of average place cell scores and reference memory errors. $r$ and $p$ denote Spearman's rank correlation coefficient and significance score, respectively. (g) place cell score distribution and relationship of average place cell scores and reference memory errors in common activation functions: GELU, ReLU, LeakyReLU, tanh, and sigmoid. All results are evaluated using training maps. (h) Rate maps of neurons with top-16 place cell scores in the FFNs with varying values of $\alpha$; $\alpha = 10$ (left), $\alpha = 1$ (middle), and $\alpha = 0$ (right).

locations, similar to the approach described in the TEM-t model [12]. We recorded the activation values of each neuron at every step during a random walk process. As the agent traversed the environment, which had an $11 \times 11$ grid structure, we accumulated the activation values for each grid point, resulting in a 2D array known as the *rate map* or *place field* (i.e., the spatial distribution of single neuron activation during a random walk, as shown in Fig.4a and Fig.4b).

Fig. 4a and 4b show the rate maps of neurons with place cell scores in the FFN and self-attention layers, respectively. For the self-attention layer, the total number of neurons in the softmax layer was 65 (context length + masked sensory observation) $\times$ 8 (number of heads) $\times$ 2 (number of layers). The total number of neurons in the FFN layer was set at 2,048 (number of neurons) $\times$ 2 (number of layers). As can be seen, our metric accurately represents location specificity. Fig. 4c and 4d show the distribution of place cell scores in the two layers with different values of $\alpha$. When the $\alpha$ value is increased, the place cell score distribution found in the FFN becomes positively shifted (see Fig. 4h rate map examples for $\alpha = 0$, $1.0$, and $10.0$), whereas the place cell score distribution in the self-attention layers remains.

Fig. 4e and 4f show a relationship between the average place cell score and the reference memory error for each $\alpha$. The average place cell scores in the self-attention layers show no correlation with reference memory errors, while neurons in the FFN exhibit a clear correlation. These results imply that reference memory formation and place cell emergence can be enhanced by NMDAR-like nonlinearity in the FFN.

Fig. 4g compares the place cell representations of our NMDA ($\alpha = 0, 10$) with the representations in FFNs with the activation functions used in Fig. 3c, indicating that the case of NMDA$_{\alpha=10}$ outperforms other activation functions, in both reference memory formation and place cell representation. Our finding that large $\alpha$ (the [Mg$^{2+}$] component) enhances reference memory is in line with the biological observation that increasing the [Mg$^{2+}$] in the brain enhances long-term memory formation [6].

We also explored the possibility of extending the hyper-parameter space by contrasting the NMDA-inspired activation function with an alternative activation function based on $\alpha$. LeakyReLU was considered for this experiment because of its adjustable negative slope, which was re-defined as $(\max(0, x) + \alpha \min(0, x))$. The results reported in Fig. S10 of Appendix A.7 show that LeakyReLU activation, for all tested $\alpha$ values, produced lower place cell scores than the best-performing NMDA-inspired function (i.e., NMDA$_{\alpha=10}$). Our findings suggest that NMDA-like nonlinearity, which is already known to be critical in biological systems, could be used to enhance transformer models.

Building upon these results, our work connects memory consolidation theory from neuroscience to transformer-based processing of short-term and long-term memory. Unlike previous studies that directly provide 2D geometric information to models [11, 10], our model only takes sensory-action pairs as an input, which was sufficient to observe the emergence of place cells in the transformer model. Specifically, we showed that nonlinearity in the transformer's FFNs plays a crucial role in transforming short-term working memory from the self-attention layer into long-term memory, and the place cell emergence. The observed place cells may correspond to sparse interpretable activation in FFN. See Appendix A.10 for further analysis on the sparsity and place cells. These findings provide new insight into the inner workings of transformers and their connection to neural processes in the brain.

## 4 Related works

The current study is inspired by recent studies that connect neuroscience and AI models. One such seminal work is by Whittington et al. [12], which showed a relationship between the self-attention layer and a recent hippocampal model called the Tolman-Eichenbaum Machine (TEM; Whittington et al. [14]). Our work expands the literature by focusing on FFNs in the transformer and making a connection to the emergence of place cells.

TEM is a neuroscience-based model that reproduces neural representations in the hippocampus and entorhinal cortex. Instead of storing memory in the key matrix $K$ and value matrix $V$, it instead stores memory using a Hebbian weight matrix $M \in \mathbb{R}^{d_k \times d_k}$. Every outer product of key and value vector $\mathbf{k}_i^\top \mathbf{v}_i$ at each step $i$ is simply stored in $M$ via the Hebbian update rule. $M$ is initialized to a zero matrix at the beginning of the task and adds every outer product at each time step:

$$M = a \sum_{i=1}^{t} \mathbf{k}_i^\top \mathbf{v}_i = a K^\top V, \tag{8}$$

where $a$ is a weighting factor. In the memory retrieving phase with the query vector $\mathbf{q}$, TEM uses an attractor network:

$$\mathbf{q} M = a \mathbf{q} K^\top V. \tag{9}$$

Whittington et al. [12] found that the memory retrieving process in TEM has a close mathematical structure to Eq. (3) when the softmax function is replaced with a linear function. We note that methods for linearizing softmax in self-attention layers have been studied to address high computational costs when context length $c$ is very long [27, 28]. The subsequent model of TEM, called TEM-t [12], replaces the attractor network (Eq. (9)) with a self-attention mechanism (Eq. (3)). This study demonstrated that TEM-t learns significantly faster than TEM.

TEM-t and TEM do not have a fixed context length $c$; therefore, these models store all information before step $t$, i.e., $c = t$. The computational cost of the self-attention layer in TEM-t is $O(t^2)$, and retaining all previous information is too expensive from both a biological and computational standpoint[3]. For TEM, the Hebbian update rule has no quadratic computational cost and can add all previous information in a fixed number of synapses $d_k^2$; however, the memory capacity of the Hebbian matrix $M$ is $O(d_k)$ and the speed of memory retrieval is substantially slower than the

---

[3]Due to the computational cost, TEM-t does not store all historical data but selectively chooses which data to store in $K$ and $V$.

self-attention mechanism [29–31]. In contrast to TEM and TEM-t that rely on a single memory system, we investigated two separate memory systems: 1) context-dependent matrices $K$ and $V$ in the self-attention layer with a fixed context length $c$ and 2) context-independent fixed matrices $U_1$ and $U_2$ (in Eq. (4)) in the FFNs.

Our research differs from previous studies in the following ways: 1) We designed a navigation test that assesses working memory and reference memory separately, providing a more comprehensive evaluation of the model's performance. 2) We proposed a new brain-inspired activation function, $NMDA_\alpha$, which relates to modern nonlinear activation functions and allows for the analysis of the effect of $\alpha$ on long-term reference memory formation. 3) We demonstrated that place cell-like neurons emerge in FFNs in conjunction with reference memory formation, which is a novel finding that has not been addressed in the TEM or TEM-t models. 4) TEM and TEM-t focus on working memory errors and do not cover reference memory errors. In contrast, our work evaluates both types of memory errors, providing a more detailed analysis of the model's performance.

## 5  Discussion

While extensive efforts have been directed toward finding the optimal nonlinear activation function for improving modern deep neural network models [21–23], their relationship to neural substrates that mediate nonlinearity in the human brain remains obscure. Furthermore, the role of nonlinearity in intelligent functions remain unclear. Our research attempts to fill this gap by proposing and testing how a biologically inspired nonlinearity functions in a transformer model that was previously related to the hippocampal formation. We examined functions in terms of long-term reference memory formation and place cell representation. This idea was evaluated in a carefully designed 2D grid environment and by implementing an activation function derived from NMDAR-like nonlinearity.

Our research reveals that place cell-like neurons that are critical to spatial navigation can be found in both the self-attention layers and FFNs of the transformer model, similar to the presence of place cells in the CA3 and CA1 regions of the hippocampus [32]. In the hippocampus, the CA3 region is thought to be involved in the initial formation of new memories, specifically in pattern completion [33], while CA1 is thought to be important for the long-term consolidation of memories. As such, we suggest that the CA3 region may serve a similar function to the self-attention layer, while the CA1 region may function similarly to FFN. However, further research is needed to fully understand the similarities and differences between the properties of place cells in the transformer model and those found in the CA3 and CA1 regions of the hippocampus. For more discussions regarding the biological plausibility of this proposal, see Appendix A.8.

Recent machine learning research has tested whether the transformer architecture is analogous to different types of biological memory. It has been suggested that (1) transformer FFN modules resemble associative memory [34], (2) the FFN in a transformer block functioning as a key-value memory [35, 36], (3) activation sparsity in the transformer FFN enhances robustness to noisy input [37, 38], and (4) sparse activity in FFNs increases the percentage of neurons that selectively activate to human interpretable input features [39].

Our data instead suggests the transformer architecture resembles *memory consolidation* by the animal brain, which refers to the transfer process of a short-term memory into a long-term memory system in neuroscience research [40]. Previous research has revealed that $Mg^{2+}$-gating of NMDA receptors modulates the formation of long-term memories [5, 6]. These observations imply that the nonlinear dynamics of NMDA receptors in hippocampus CA1 are critical for consolidating short-term memory into long-term memory.

To our surprise, our results agree qualitatively with previous NMDAR impairment experiments from neuroscience: 1) selective inhibition of hippocampal CA1 NMDAR inhibition does not disrupt working memory [41] but impairs the long-term memory formation [2], 2) changing NMDAR $Mg^{2+}$-gating (changing $\alpha$ in this work) enhances or disrupts long-term memory formation [5, 6], 3) NMDAR is required for long-term stabilization of newly forming place fields [9, 3]. These similarities between hippocampal memory consolidation and our results suggest that the transformer is an effective memory consolidation model.

Our research points to exciting future directions. The current study examined what-where memory using a sensory observation task in a static environment. However, our real-world environment

changes constantly and provides new inputs over time. Modern deep learning systems are generally incapable of adapting to a dynamic environment or reordering of sensory inputs. We intend to explore what-where-when memory, called *episodic memory*, in transformers and other deep models.

## Acknowledgments and Disclosure of Funding

This work was supported by the Institute for Basic Science in Korea (IBS-R029-C2, IBS-R001-D2). We sincerely thank Dr. Sungho Hong of the Okinawa Institute of Science and Technology for his thoughtful feedback on computational neuron modeling.

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

# A   Appendix

## A.1   Place cell

### A.1.1   Place cell overview

Place cells are a distinct class of neurons in the hippocampus that fire selectively in a certain location [S1], and provide a sparse population code for self-location. Place cells play a key role in understanding how the hippocampus represents spatial locations [S2], which is characterized by their place fields, which are spatially stable regions where the cell preferentially fires. As the subject moves through the environment, different place cells fire in response to its position, collectively forming a cognitive map [S3]. This dynamic encoding allows the brain to create and maintain spatial representations, which are essential for navigation and memory [S4].

**Relevance to machine learning**   Accurately identifying place cells and understanding their properties is crucial for machine learning practitioners working on spatial representation, navigation, and memory problems. Recent studies investigated the role of place cells and grid cells in spatial representation in navigation tasks with deep models [S5–S8]. The concept of place cells can provide insights that support the development of novel algorithms and techniques for potential machine learning applications such as robotics, autonomous vehicles, virtual reality, and other areas where spatial understanding is essential.

**Grid cells and path integration**   While our work was mainly on the emergence of place cells, we could also observe grid cells based on the experiment, similar to TEM and TEM-t models. When we removed the recurrent connection in the action embedding layer, grid cells did not emerge, and the model's performance declined, indicating the role of grid cells in place cell formation and spatial memory. An interesting hypothesis in neuroscience is that the interaction between grid cells, path integration, and place cells forms a critical loop [S9, S10] in the neural circuitry underlying spatial memory formation, which is similar to our findings from removing recurrent connections. This finding awaits future investigation on how grid cells and path integration affect place cell emergence and spatial memory formation.

### A.1.2   Place cell score calculation

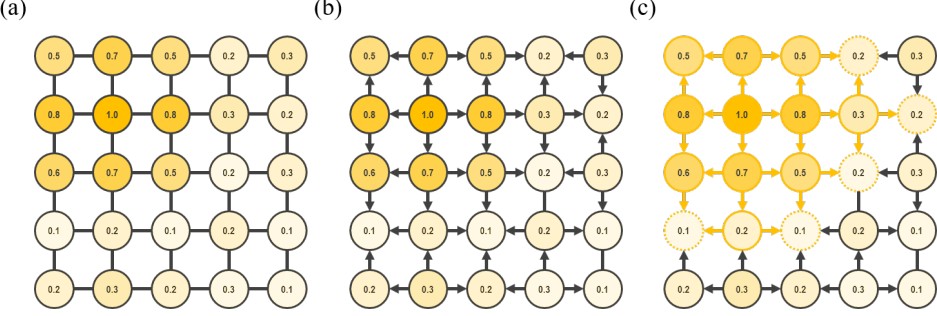

Figure S1: Schematic diagram for creating auxiliary graph for place cell score calculation (a) Consider the 2D grid environment of the place field as an undirected graph $\mathcal{G}$. Colors represent the firing rate of nodes. (b) Convert $\mathcal{G}$ into a directed graph by comparing the firing rate $\rho$ between nodes (direction of higher to lower firing rates) (c) Operate Depth First Search algorithm from peak node to construct all connected components }(yellow). Dashed circles are leaf nodes.

Our place cell score metric is inspired by the metric described in the TEM-t model [S8]. The place cell score in Eq. S1 is a scalar value that quantifies each neuron's firing specificity for a grid location. We record every neuron's activation value at every step during the random walk process. Since the map has an $11 \times 11$ grid structure, we accumulate the activation value for each grid point as the agent random walks, creating a 2D array. This array is called the rate map or the place field (i.e., the spatial distribution of single neuron activation during a random walk).

We investigated the role of neurons in the FFNs and self-attention layers by measuring their place specificity. Given a $K \times K$ 2D grid environment as graph $G = (V, E)$ and a firing rate (i.e., cumulative activation value at node $i$ divided by the length of evaluation trial) of node $i \in V$ as a $\rho_i$, we defined a *maximally firing node* as $i_{\max}$ and its firing rate as $\rho_{\max}$. $E$ are directed edges that connect high to low firing nodes in $G$. From $G$, we ran a depth-first-search from source node $i_{\max}$ to build a sub-graph $\mathcal{G} = (\mathcal{V}, \mathcal{E})$ which we call *all connected components* S1. Given $G$ and $\mathcal{G}$, the *place cell score* is defined as follows:

$$\text{Place cell score} = \gamma \frac{\sum_{i \in \mathcal{V}} \rho_i}{\sum_{i \in V} \rho_i}, \tag{S1}$$

where $\gamma = 1 - |\mathcal{V}^*|/|V|$ is a discount factor and $\mathcal{V}^*$ is $\mathcal{V}$ without node $i_{\max}$ and leaf nodes. To measure place cell score, we recorded the firing rate $\rho_i$ of neurons over a random walking trajectory with $10^5$ steps in one of the training maps. We then measured the place cell scores of neurons in the FFN and self-attention layers. The place cell score is 1 when the neuron is firing only at a specific node; the score is 0 when it fires homogeneously across all nodes.

Since 2D-grid spaces cannot be directly interpreted as Euclidean spaces, it is not appropriate to apply space-related functions, such as Gaussian kernel density estimation, to the place fields. Place cell evaluation methods in previous research can be broadly categorized into four types [S11]:

1. **Peak method:** Classifies cells based on the average firing rate in one location being higher than in the rest of the environment.

2. **Stability method:** Classifies cells with stable firing patterns across locations over time.

3. **Information method:** Classifies cells based on the increased amount of spatial information they hold about the animal's location.

4. **Combination method:** A combined approach that considers cell's place fields, including their size, peak, and activity.

We chose the Peak method approach because it easily applies to 2D-grid structures.

To implement the Peak method in a 2D grid environment, we represent the 2D grid environment as a graph structure where neuron activities at each node are tracked while the agent explores the environment. Averaging neuronal activities per node yields the neuron's place fields, as illustrated in Figures 4 and 5. In these place fields, we find a node with a maximal firing rate, $i_{\max}$ Starting from this node, we find all connected components as a subgraph by connecting neighboring nodes within the nearest minimum. We create directed edges connecting nodes from high to low firing rates and run depth first search (DFS) from the node $i_{\max}$. DFS can be considered an inverted watershed algorithm that fills a single pool containing the global minimum. The advantage of our place cell metric is its ability to generalize to various graph structures, such as 2D hexagonal grids or 3D grid spaces.

### A.1.3 Pseudo code for calculating place cell score metric

function PlaceCellScore(place field)
**Input** : place field ($K \times K$ 2D array)
**Output** : place score
$G$ := 2D grid graph ($K \times K$)
$\mathcal{G}$ := empty directed graph ($K \times K$)
**for** *edge* ($node_i \rightarrow node_j$) *in* **G do**
    **if** *firing rate $\rho_i$ > firing rate $\rho_j$* **then**
       |   $\mathcal{G}$ add *edge* ($node_i \rightarrow node_j$)
    **end**
**end**
Find $node_k$ of firing rate $\rho_{max}$
**for** $node_v$ *in* $\mathcal{G}$ **do**
    **if** $node_v$ *is not descendant of $node_k$ found with DFS(k)* **then**
       |   delete $node_v$ from $\mathcal{G}$
    **end**
**end**
conn. components = sum of all nodes' firing rates in $\mathcal{G}$
total components = sum of all nodes' firing rates in $G$
place score = $\gamma \dfrac{\text{conn. components}}{\text{total components}}$
$^\dagger$ $\gamma$ is discount factor, determined by connected component size
**return** place score

**Algorithm 1:** Pseudo code for calculating place cell score metric

The place field in Algorithm 1 is measured as following procedure:

1. During a random walk simulation, the activation value of a neuron at node $i$, where the agent is located, is measured every 65 steps. Let's say this value is $a_i$.

2. Every time the agent visits node $i$ again, value $a_i$ is added cumulatively to the recorded value; $A_i += a_i$ such that $A_i$ is the cumulative activation value at node $i$. We assume the initial value of $A_i$ is zero.

3. After the random walk is done, $A_i$ divided by the length of the random walk trajectory is the firing rate $\rho_i$ at node $i$ of the neuron (place field $\in \mathbb{R}^{K \times K}$).

In our place cell evaluation experiment, the length of the random walk is $10^5$ and $K = 11$; the evaluate map is one of the training maps.

### A.2 Derivation of NMDAR nonlinearity from the molecular level chemical interaction

Here, we describe the NMDAR nonlinear dynamics from chemical interaction between $Mg^{2+}$ and NMDAR following previous literature [S12–S14]. At the molecular level, one $Mg^{2+}$ ion binds to one NMDAR receptor when opening the NMDAR channel. Thus, the chemical equation of binding reaction between $Mg^{2+}$ ion and NMDAR receptor, R, can be described as

$$Mg^{2+} + R \rightleftharpoons Mg^{2+}R. \tag{S2}$$

From this chemical equation, the equilibrium constant $K$ is given by

$$K = \frac{[Mg^{2+}R]}{[Mg^{2+}][R]}. \tag{S3}$$

Thus, dissociation constant $K_D$, which correspond to $Mg^{2+}$ dissociation from NMDAR, follows

$$K_D = K^{-1} = \frac{[Mg^{2+}][R]}{[Mg^{2+}R]}, \tag{S4}$$

in which [R] and [$Mg^{2+}$R] are the free and $Mg^{2+}$-bound NMDARs respectively. The fraction of opened NMDAR channels (number of unbound NMDAR over a number of total NMDAR) at equilibrium follows,

$$\begin{aligned}
\mathbf{p} &= \frac{[\text{R}]}{[\text{R}] + [\text{Mg}^{2+}\text{R}]} \\
&= \frac{1}{1 + [\text{Mg}^{2+}]/K_D}
\end{aligned} \tag{S5}$$

Experimentally, the voltage-dependent dynamics of $K_D$ has been described as following equation by Ascher and Nowak [S15]

$$K_D = K_{\text{Mg}^{2+}} e^{\beta V}, \tag{S6}$$

where, $V$ is membrane voltage, $\beta$ is a temperature constant and $K_{\text{Mg}^{2+}}$ is a dissociation constant at $V = 0$. If Eq. S6 is substituted into Eq. S5, voltage-dependent open fraction of NMDAR can be expressed as follows:

$$\begin{aligned}
\mathbf{p}(V) &= \frac{1}{1 + \frac{[\text{Mg}^{2+}]}{K_{\text{Mg}^{2+}}} e^{-\beta V}} \cdot \\
&= \frac{1}{1 + \alpha e^{-\beta V}} \cdot
\end{aligned} \tag{S7}$$

in which $\alpha = [\text{Mg}^{2+}]/K_{\text{Mg}^{2+}}$, the parameter determined by the $[\text{Mg}^{2+}]$. Given the voltage-dependent open fraction of NMDAR, $\mathbf{p}(V)$, and NMDAR's maximal conductance, $g_{\text{max}}$, then voltage-dependent NMDAR conductance $g(V)$ can be described as

$$g(V) = g_{\text{max}}\mathbf{p}(V) \tag{S8}$$

Given $g(V)$, and driving force, $V - V_r$, and current $I$, they have a relationship of $I = (V - V_r)g(V)$, in which $V_r$ is reversal potential (the value of membrane potential above which current inverts the direction). As experimental investigations on the physiological reversal potential of NMDAR to be $V_r = 0$ [S16–S18], $I = Vg(V)$. Then, the normalized NMDAR current $I_{\text{norm}} = I/g_{\text{max}}$ follows:

$$I_{\text{norm}} = V\mathbf{p}(V) \tag{S9}$$

From Eq. S9 and previous electrophysiological experimental results [S19], we reconstruct IV curve in (Fig. 1a, right top).

### A.3 NMDAR-inspired nonlinear activation function

Here, we propose an NMDAR-inspired nonlinear activation function from the nonlinear dynamics of the NMDAR-IV curve. If we consider the nonlinear IV curve of NMDAR (Eq. S9) as a nonlinear mapping function, $\phi$, the membrane voltage, $V$, can be viewed as an input, $x$, and normalized NMDAR current, $I_{\text{norm}}$, as an output, $\phi(x)$. Therefore, we can rewrite the nonlinear mapping function $\phi$ as follows

$$\phi(x) = x\mathbf{p}(x). \tag{S10}$$

We define the NMDAR-inspired activation function as a nonlinear mapping function, $\text{NMDA}(x) := \phi(x)$. By substituting Eq. S7 into Eq. S10, we show the generalized expression of $\text{NMDA}(x)$ equation with $\alpha$ and $\beta$ parameters as follows:

$$\begin{aligned}
\text{NMDA}_{\alpha,\beta}(x) &= x\mathbf{p}_{\alpha,\beta}(x) \\
&= \frac{x}{1 + \alpha e^{-\beta x}}.
\end{aligned} \tag{S11}$$

Given $\alpha = 1$ and $\beta = 1$, $\mathbf{p}(x)$ is identical to the sigmoid function, $\sigma(x) = 1/(1 + e^{-x})$. This particular case of $\alpha$ and $\beta$ leads to $x\sigma(x)$, Sigmoid Linear Unit (SiLU) activation function [S20]. In the case of $\alpha = 1$ and $\beta = 1.702$, $x\sigma(1.702x)$ corresponds to the GELU activation function, which is popular in transformer models [S21, S22, S21]. Ramachandran et al. [S23] introduced the swish activation function, $x\sigma(\beta x)$, which is a generalized form of GELU and SiLU. They demonstrated

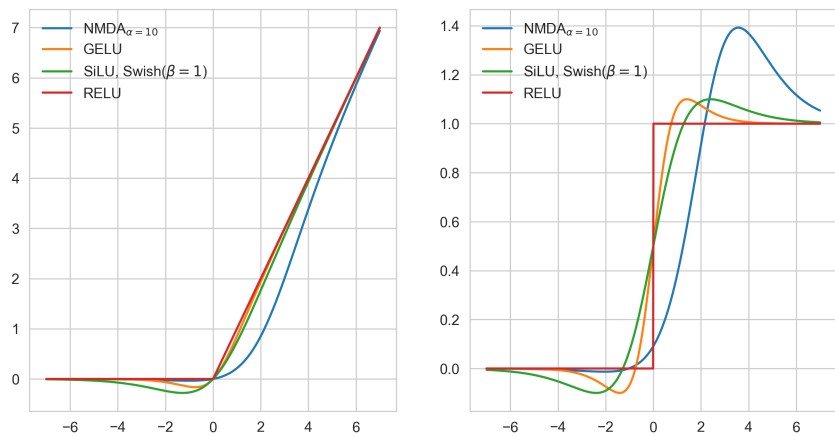

Figure S2: Comparison of common activation functions (left) and their derivatives (right) with $\text{NMDA}_{\alpha,\beta}$.

that when $\beta \to \infty$, the activation function resembles RELU. We summarized these four activation functions by comparing them with our $\text{NMDA}_{\alpha,\beta}(x)$ in Table 1 and Fig. S2.

In contrast to the extensive research on $\beta$ in $\text{NMDA}_{\alpha,\beta}(x)$, $\alpha$, the $\text{Mg}^{2+}$-gating component, is not explored. For this reason, we focused on the parameter $\alpha$ over $\beta$, and investigated $\text{NMDA}_{\alpha}(\text{X})$. It is interesting to note that the Swish function was originally proposed as a self-gating function, inspired by the use of the sigmoid function as a gating of information flow in the long short-term memory (LSTM) network [S24]. In contrast, our activation function $\text{NMDA}(x)$ is inspired by the physical $\text{Mg}^{2+}$-gating mechanism that occurs at the real biological synapses. These shared mechanisms of self-gating in artificial models and biological observations raise the interesting possibility that NMDAR is a neural substrate of nonlinear activation function in the brain.

## A.4  Detailed description of task design and definition of short-term working memory and long-term reference memory

Our task is based on a widely employed neuroscience experiment for two separate memory systems: working memory and reference memory [S25, S26]. Errors in working memory are measured by within-trial error, whereas errors in reference memory are measured by across-trial error. The training phase and the test phase alternate at each trial. In the test phase, the unvisited place prediction error and visited place prediction error for the familiar map and the novel map, respectively, are measured. The memory of a relatively recent experience can be defined as *short-term working memory* (STWM), and the memory of relatively old experience can be defined as *long-term reference memory* (LTRM). Within trial visited place prediction measures relatively short-term experience for our task. On the other hand, the across-trial unvisited place prediction task in the familiar map measures the relatively long-term experience. Measuring unvisited place prediction error in the novel map will establish a baseline of chance-level accuracy; above this baseline, the formation of long-term memory can be observed (Fig. S3).

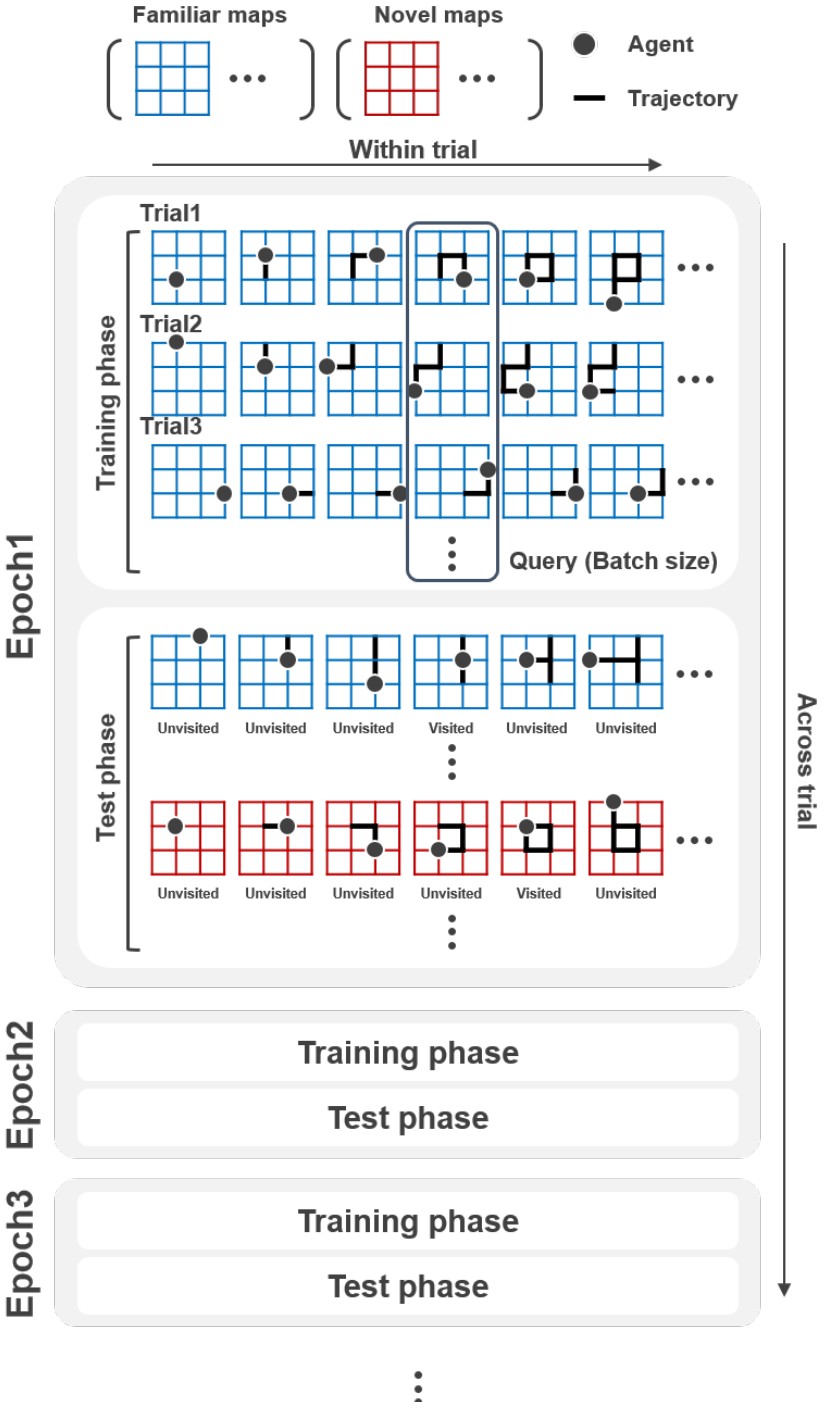

Figure S3: Detailed task design of working and reference memory evaluation. At each random walk step, a batch is created (which is then used in the backpropagation step). The batch size is 512 since there are 512 parallel random walkers in use. Note that at each trial the agent randomly selects a map from training maps (familiar maps), the initial position of the agent is random, and the agent does a random walk.

## A.5 Non-recurrent positional embeddings and prediction errors on the node visited for the first time

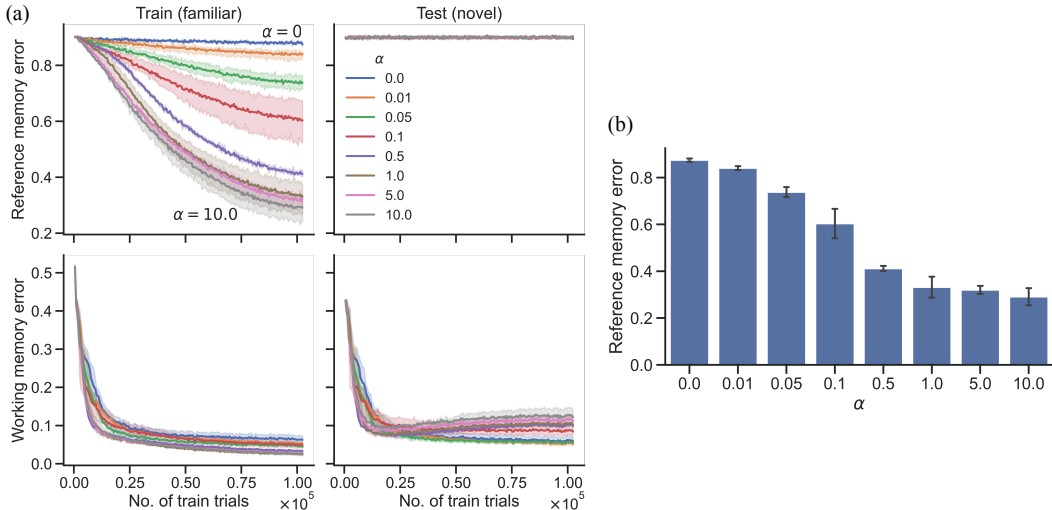

Figure S4: Experiment with non-recurrent positional embeddings. (a) Reference and working memory errors over training trials for training (familiar) maps and testing (novel) maps for $N = 32$ where $N$ is the number of training maps. (b) Reference memory errors evaluated on training maps over different values of $\alpha$ in $\text{NMDA}_\alpha$ for $N = 32$. Error bars and shaded areas represent the standard deviation of errors from three independently trained models.

**Non-recurrent positional embeddings**    We tested the non-recurrent positional embedding by substituting the recurrent positional embedding $\mathbf{e}_t$ with the action embedding $A(a_t)$, where $A$ is the embedding layer and $a_t$ is the action at step $t$. Compared to Fig. S4a, the result demonstrates a significant increase in working memory error and reference memory error (Fig. 3 vs. Fig. S4). Nonetheless, the model's behavior is comparable to the trend of decreasing reference memory error with increasing $\alpha$ of $\text{NMDA}_\alpha$ (see Fig. S4b).

Moreover, we found that place cells do emerge in the feedforward layers for this standard positional embedding method (see Fig. S5). It is worth noting that these place cell scores are noticeably lower compared to those obtained through the recurrent positional embedding. Despite the hindered performance in working memory and reference memory, a strong correlation persists between the place cell scores in FFNs and the reference memory errors (see Fig. S5c and S5d).

**Prediction errors on the node visited for the first time**    We compared unvisited node prediction error (unvisited within context window, in this case, 64 steps) versus first visited node prediction error (unvisited for within a trial). As shown in Fig. S6, the prediction error results for the first visited node do not differ from the reference memory error results.

These results strongly support the conclusions that (1) while the path-integrated information from recurrent positional embedding is important for learning the spatial structure of the map, this information is not used in predicting the unvisited node, and (2) the reference memory is used for predicting an unvisited node on a familiar map.

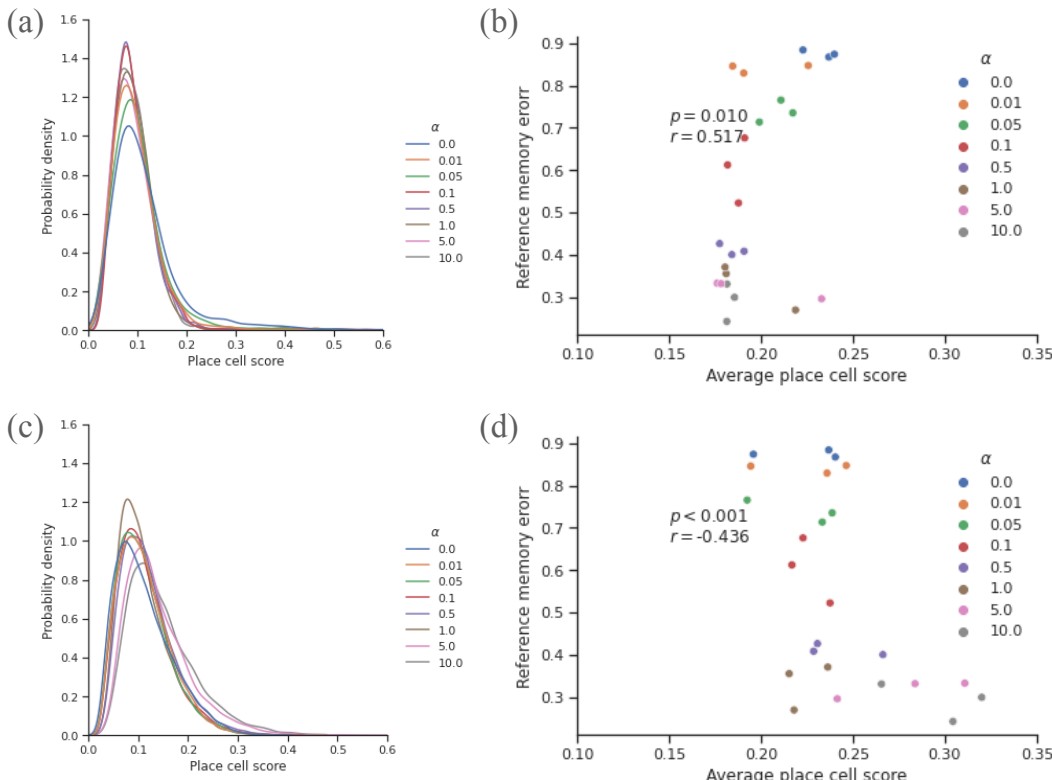

Figure S5: Place cell score analysis for non-recurrent positional embeddings for $N = 32$. Place cell score distributions with varying $\alpha$ in self-attention layers (a) and feed-forward layers (c). Scatter plot of average place cell scores and reference memory errors for self-attention layers (b) and feed-forward layers (d). $r$ and $p$ denote Spearman's rank correlation coefficient and significance score, respectively.

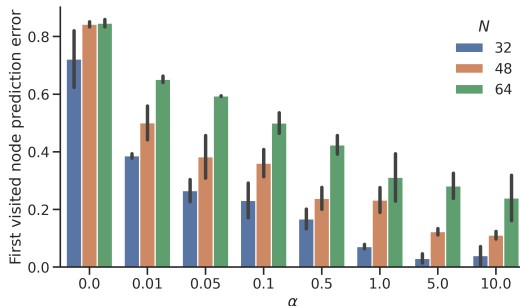

Figure S6: First visited node prediction error evaluated on training maps over different values of $\alpha$ in NMDA$_\alpha$ for $N = 32, 48$, and $64$. Error bars and shaded areas represent the standard deviation of errors from three independently trained models.

## A.6 Statistical significance between the NMDA and GELU activation functions & learning rate effect: is it an under-trained effect?

In this section, we address concerns about the NMDA style activation with large $\alpha$ being a trivial effect for the emergence of place cells and the improvement of learning speed. We provide additional evidence to support our claims and demonstrate that the observed effects are not trivial or due to under-training.

**Statistical test across activation functions**    Our primary claim regarding the effectiveness of NMDA-style activation is based on the results shown in Fig. 3. To support our claim, we conducted a

statistical across various activation functions. Our findings revealed a significant difference between $NMDA_\alpha$ and other activation functions, indicating that the NMDA style activation is beneficial for the emergence of place cells and the improvement of learning speed (see Table S7).

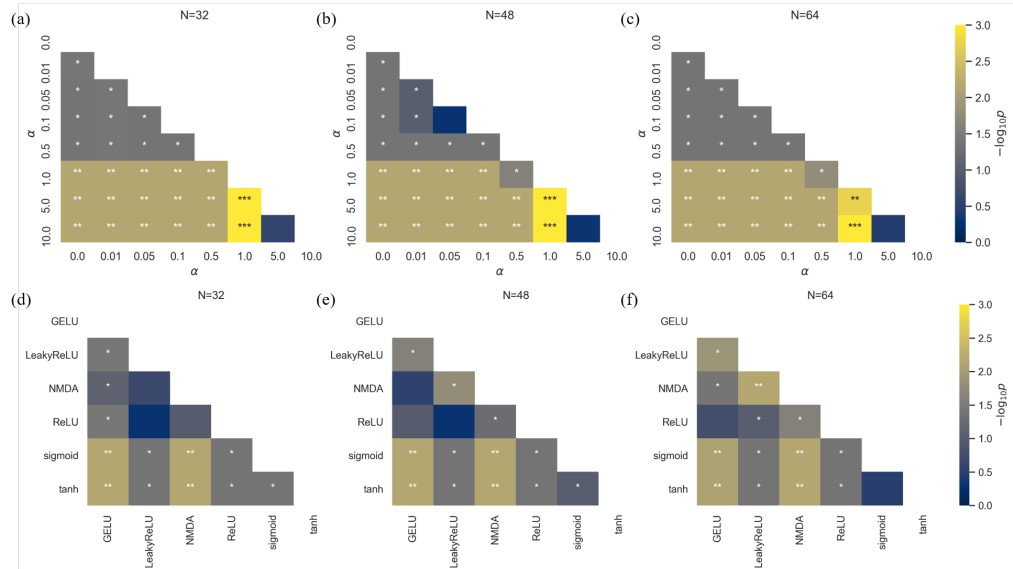

Figure S7: Statistical test result across various activation functions with different numbers of maps (N) related to Figure 3. (a-c) Mann-Whitney test across various $\alpha$ of $NMDA_\alpha$. (d-f) Mann-Whitney test across various activation functions. The sample size (or trials) used for tests ranges from 3 to 11. Colors represent $p$-values and asterisks indicates statistical significances: * 0.01<p<0.1; ** 0.001<p<0.01; *** p<0.001. $\alpha = 10$ for NMDA and $\alpha = 0.01$ for LeakyReLU.

**Gradient around** $x = 0$ **and learning speed**     The performance gain at large alpha values might be due to an increased gradient around $x = 0$. However, the gradient around $x = 0$ at $\alpha = 1$ (green) is actually larger than $\alpha = 10$ (blue) in Fig. S2. Therefore, the performance gain at large alpha values is not due to an increased effective learning speed around $x = 0$. Instead, it may be attributed to the activation function's ability to better capture nonlinear relationships.

**Learning rate effect on reference memory error**     To clarify that our result is not simply due to the under-trained model, we performed experiments with varying learning rates: 0.001, 0.0005, 0.0003, 0.0002, and 0.0001. As can be seen in Fig. S8, a larger learning rate at low alpha values is not helpful for reducing reference memory error. It is worth noting that all models' working memory error is almost zero. Larger learning rates lead to faster decrease in working memory error, but at high learning rates, reference memory error stays around the chance level of 0.9, i.e., no reference memory emerges. Additionally, Fig. S9 shows that higher place cell scores correspond to lower reference memory errors. Note that we employed a linear decay learning rate schedule (from the starting learning rate to 0) and the starting learning rate used in our manuscript is 0.0001.

In conclusion, our additional experiments and analyses demonstrate that the NMDA style activation with large $\alpha$ plays a critical role in the emergence of place cells and the improvement of reference memory, and it is not a trivial effect or an under-trained effect.

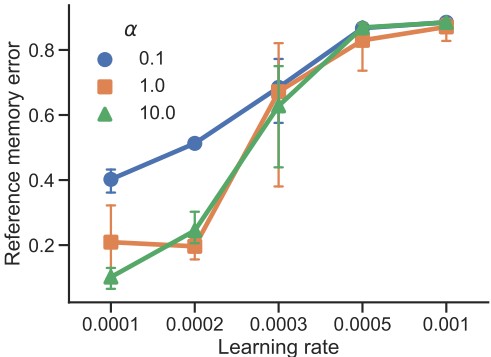

Figure S8: Familiar map reference memory error results with five different learning rates, where $\alpha \in \{0.1, 1.0, 10.0\}$, for a total of 64 training maps ($N = 64$). The error bars represent the standard deviation of errors from three independently trained models.

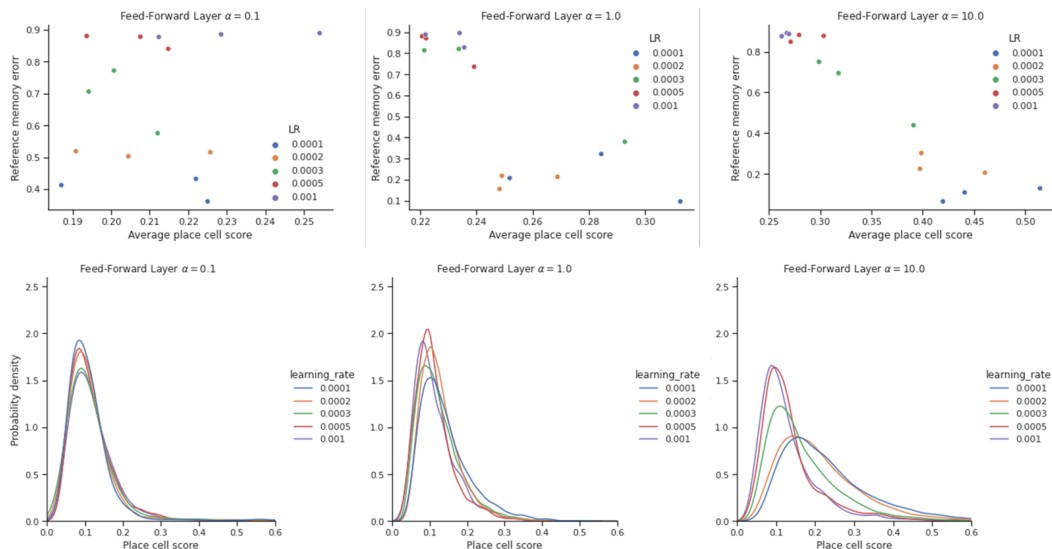

Figure S9: (top) Scatter plot between reference memory error and average place score; (bottom) Distribution of place cell scores with five different learning rates, where $\alpha \in \{0.1, 1.0, 10.0\}$ (shown at the top of each figure), for a total of 64 training maps ($N = 64$).

## A.7 Consequent of changing nonlinear dynamics in Leaky ReLU activation function

Here, we investigated the consequence of changing nonlinearity with an activation function other than NMDA$_\alpha$. We chose LeakyReLU ($\max(0, x) + \alpha \min(0, x)$) activation function to compare with NMDA$_\alpha$. Regarding LeakyReLU, $\alpha = 1$ of LeakyReLU also leads to a fully linear activation function similar to $\alpha = 0$ of NMDA$_\alpha$. Compared to NMDA$_{\alpha=10}$, LeakyReLU exhibits a lower average place score in the allowed range of $\alpha$, indicating that NMDA$_\alpha$ performs better in terms of robust place cell emergence (see Fig. S10)

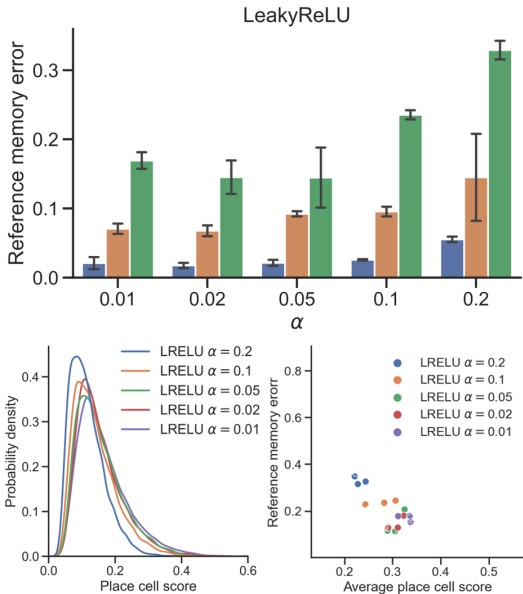

Figure S10: Evaluation of reference memory error in LeakyReLU (LRELU) while modulating $\alpha$(top) and relationship of average place cell score and reference memory error (bottom).

## A.8 Transformer as a memory consolidation model and its biological plausibility

Next, we examined the biologically inspired NMDA$_\alpha$ activation function in the feed-forward layer of the transformer and its role in memory formation and place cell representation. We show that modulating $\alpha$ corresponds to a change in extracellular [Mg$^{2+}$], by deriving the nonlinear activation function from the real NMDAR nonlinear IV curve. The reconstructed real nonlinear IV curve is shown in Fig. 1a (top right panel).

The modulation of $\alpha$ selectively affects the formation of long-term reference memory (i.e., prediction of unvisited places across trials) while leaving the formation of short-term working memory (i.e., prediction of unvisited places within trials) intact. This result suggests that short-term working memory and long-term reference memory are physically stored in separate structures: the self-attention layer and the feed-forward layer. In psychology, a multi-store model for the animal brain that comprises short-term memory and long-term memory was historically suggested in [S27]. In this biological model, sensory inputs are stored in short-term memory systems via attention, and some are transferred to a long-term memory system while others quickly disappear.

In neuroscience, the transfer of short-term memory into a long-term system is called *memory consolidation* [S28]. Animal studies have demonstrated that the CA1 region of the hippocampus is essential for memory consolidation [S29, S30]. In hippocampal CA1, the postsynaptic NMDA receptor mediates synaptic plasticity, and the selective perturbation of these receptors leads to impairment in long-term memory formation [S31, S30]. Later research revealed that Mg$^{2+}$-gating of NMDA receptors modulates the formation of long-term memory [S32, S33]. These observations imply that the nonlinear dynamics of NMDA receptors in CA1 are critical for consolidating short-term memory into long-term memory S11.

**On the basis of a previous link between the hippocampus and the transformer model, we hypothesize that the latter is a model for memory consolidation.** Given the resemblance of the GELU nonlinear activation function and CA1 NMDAR nonlinear IV curve, we assumed that the GELU activation function serves as a key component that links short-term working memory and long-term reference memory. Our experimental results indicate that that the formation of long-term reference memory is impaired when the activation function is completely linear (corresponding to no Mg$^{2+}$). In contrast, increasing $\alpha$ (which corresponds to an increase in Mg$^{2+}$ level) revealed that for long-term reference memory, our model outperforms other activation functions (e.g., RELU, GELU, LRELU, Sigmoid, Tanh). Based on these similarities between hippocampal memory consolidation and our results, we propose that transformer is an effective memory consolidation model.

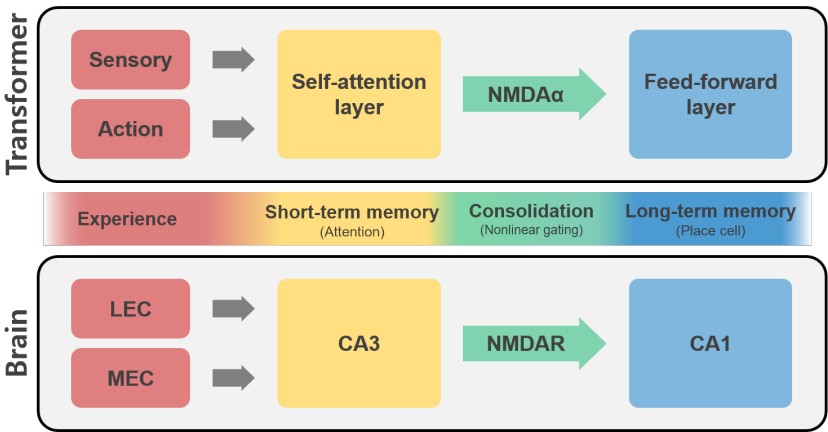

Figure S11: Schematic illustration of the transformer as a memory consolidation model of hippocampus via NMDAR nonlinearity. Top, transformer with $NMDA_\alpha$ nonlinear activation function.Bottom, summarized brain model with NMDA receptor mediated long-term memory formation. Colors represent the corresponding stage of the memory consolidation process.

In addition to the performance gain in long-term memory formation with $NMDA_\alpha$, we found that modulating the $\alpha$ affects the emergences of place cells in the feed-forward layer and conclude that there is a significant correlation between place cell score and long-term reference memory formation. Our results align with previous biological findings that perturbation of CA1 NMDARs leads to impairment in both place cell representation and long-term memory formation [S31, S34, S35, S29]. These similarities support the idea that place cells are the neural correlates of long-term spatial memories. Altogether, our findings suggest the exciting possibility that the nonlinear IV curve of NMDAR in the hippocampal CA1 is a neural substrate of nonlinear activation function in the brain.

### A.9 GPT2 and Vision Transformer with NMDA activation functions

In this section, we present the detailed experimental setup and results for applying the proposed $NMDA_\alpha$ activation function to the GPT2 [S36] and Vision Transformer (ViT) models.

#### A.9.1 Language Modeling with GPT2

We replaced the GELU activation function in the feed-forward networks (FFNs) of the GPT2 model (model size: 124M) with the $NMDA_{\alpha=10}$ activation function and conducted experiments on the Open-WebText dataset [S37]. We employed the GPT2 training PyTorch code from `karpathy/nanoGPT` GitHub repository [S38]. The application of $NMDA_{\alpha=10}$ showed a slight increase in performance, as illustrated in Fig. S12.

#### A.9.2 Image Classification Tasks with ViT

We conducted experiments on image classification tasks using the ViT model, and the results were examined on the CIFAR-100 and TinyImageNet datasets. Three trials were conducted for each condition, and an increasing, though statistically insignificant, tendency in performance was observed. The top-1 test accuracies for these datasets are provided in Table S1.

Table S1: Top-1 test accuracies with ViT model on CIFAR-100 and TinyImageNet datasets.

| Dataset | GELU | $NMDA_{\alpha=10}$ | $NMDA_{\alpha=0}$ |
|---|---|---|---|
| CIFAR100 | $69.92 \pm 0.34$ | $70.27 \pm 0.47$ | $49.91 \pm 0.58$ |
| TinyImNet | $54.90 \pm 0.45$ | $55.72 \pm 0.03$ | $40.36 \pm 0.52$ |

**Hyperparameter Consideration** It is important to note that we did not scan a range of hyperparameters for either NMDA or the GELU-based model. The optimal hyperparameters for NMDA might differ from those for the GELU-based model.

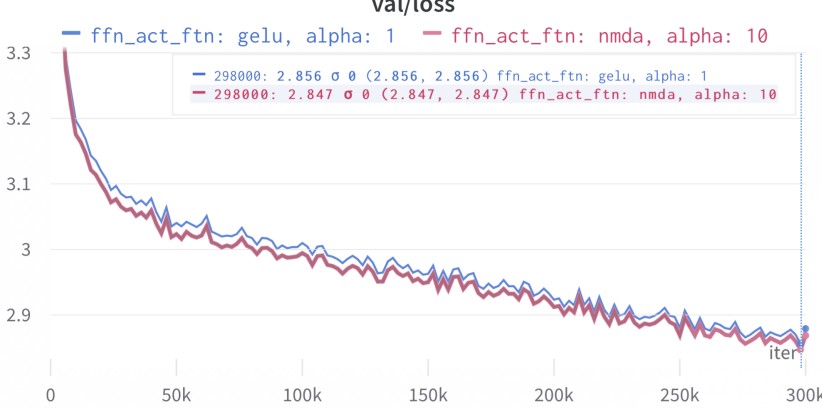

Figure S12: Validation loss curve over training iteration for GPT2 model on the OpenWebText dataset. The blue (red) line represents the GPT2 with GELU (NMDA$_{\alpha=10}$) function.

**Future Works**  Although these results are promising, they are not conclusive in stating that NMDA generally outperforms GELU for various real-world tasks. Future works will include evaluating our NMDA function on different model sizes of GPT or other language models, as well as evaluating the pre-trained models on downstream NLP tasks such as reading comprehension, question answering, common sense, MMLU, and BIG-bench.

### A.10  Sparsity of Activation

In this section, we present more intuitive evidence to support the effectiveness of our proposed nonlinearity, based on the sparsity of activation in FFNs.

**Mathematical Intuition**  Given the NMDA$_\alpha$ activation function, NMDA$_\alpha(x) = x/(1 + \alpha e^{-x})$, we can rewrite this function as follows:

$$\text{NMDA}_{\alpha(x)} = x \cdot \frac{1}{1 + e^{-(x-c)}}, \quad \text{where } \alpha = e^c. \tag{S12}$$

In the given expression, an increase in $\alpha$ results in shifting the sigmoid function towards the positive direction by increasing $c$. This shift can be understood as an increase in the threshold of information gating, which is mediated by the sigmoid function. Consequently, an increase in $\alpha$ might enhance the sparsity of the NMDA activation.

**Empirical Results**  To validate the aforementioned mathematical intuition, we conducted measurements on the sparsity of the activities in the feed-forward layer by calculating the Gini index [S39] for each input sequence. The Gini index is defined as follows:

$$G = \frac{\sum_{i=1}^{K} \sum_{j=1}^{K} |x_i - x_j|}{2K^2 \bar{x}}, \tag{S13}$$

where $x_i$ is the i-th neuron's activation value and $\bar{x} = \sum_{i=0}^{K} x_i/K$ is the mean of the activation values in the feed-forward layer ($K = 2048$ is the total number of neurons in a feed-forward layer).

The Gini index ranges from 0 to 1. When only a few neurons have very high activation values and others are small, the Gini index is close to 1; on the other hand, when most neurons have homogeneous activation values, this value is close to 0 (1 = absolute sparsity, 0 = all activations equal).

According to Fig. S13a, when the value of $\alpha$ in NMDA$_\alpha(x)$ is increased, the Gini index also increases. This increase signifies that the population activities become more heterogeneous, leading to a heavy-tailed distribution. Furthermore, the Gini index of NMDA with an $\alpha$ value of 10 is greater than that of the GELU activation function (indicated by the dashed line). These findings imply that NMDA$_\alpha(x)$ could potentially improve long-term memory formation by promoting sparsity in the activations of the feed-forward layer.

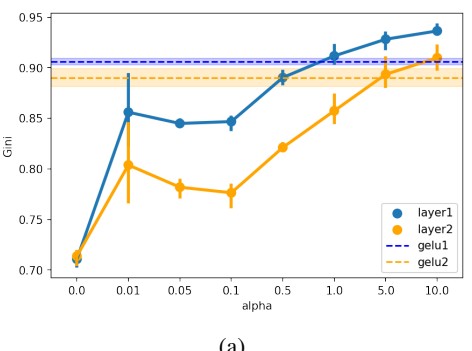
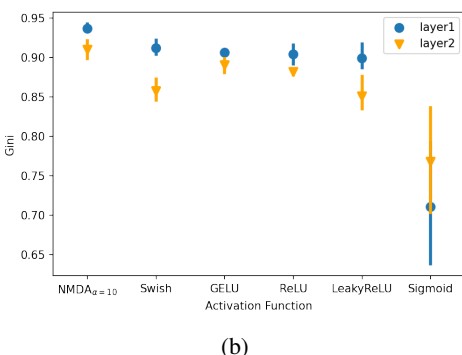

|(a)|(b)|

Figure S13: (a) Comparison of Gini index for different values of $\alpha$ in NMDA$_\alpha$. (b) Comparison of Gini index between different nonlinear activation functions.

Previous studies have explored the generalization performance and sparsity in overparameterized models [S40]. Although the mechanism behind the emergence of sparse representation in large models is not fully understood, it is worth noting that overparameterized models converge to simpler models than those with dense representations.

The high sparsity properties of the NMDA function may contribute to the increased score of place cells. In the case of Transformers, enforcing neuron activation sparsity in MLPs has been found to improve the interpretability or selectivity of a higher percentage of neurons [S41]. This evidence could potentially explain the higher place cell score observed when the alpha value is increased. We observed that NMDA with $\alpha = 10$ exhibits the highest Gini index among different activation functions. See Fig. S13a for a comparison of Gini index among activation functions.

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
