# OpenReview forum: "Transformer as a hippocampal memory consolidation model based on NMDAR-inspired nonlinearity"
_NeurIPS.cc/2023/Conference — NeurIPS 2023 poster_

### Official Review · Reviewer_FYwU · 2023-07-05

**Soundness:** 3 good
**Presentation:** 3 good
**Contribution:** 3 good
**Rating:** 7
**Confidence:** 4

**Summary:**

This paper uses a transformer on neuroscience relevant task of spatial navigation, and shows when trained on both novel and familiar tasks simultaneously, place cell representations appear is different parts of the transformer depending on the current task – place cells in the feedforward net (post self attention) for familiar tasks, and in the self-attention layer for both novel and familiar tasks. The ability for the transformer to perform on familiar tasks is related to the activation function in the feedforward net.

**Strengths:**

Interesting question of different representations for different task distributions. Gets at hippocampal consolidation which is under-explored with models. Convincing simulation results.

**Weaknesses:**

1)	The NMDAR part of the paper is the least convincing, being that standard activation functions like ReLU seem to work just as well. Don’t think you can have sentences like ‘We find that NMDAR-like nonlinearity is essential for shifting short-term working memory long-term reference memory in transformers’.
2)	The NMDAR impairment is essentially just making the activation function more linear, which of course will mean that it can’t effectively store long term memories.
3)	To effectively make the NMDAR receptor point as a different activation function class, you’d need to test it on more standard ML tasks, rather that bespoke neuroscience tasks.
4)	You have shown a nice potential neuro-ai link via NMDAR, but it’s currently just a potential relationship. It would need a lot more testing to make it concrete, e.g. showing that alpha=10 (the best performing model) is the regime that the brain operates in etc.


**Questions:**

Presumably episodic memory is a better name than working memory, being that you’re saying it’s hippocampal related.

See weaknesses for other questions.


**Limitations:**

As described in the weaknesses, the main limitation is about the interpretation of NMDAR and the relevance of the activation function for ML. Otherwise the paper is good.

---

> ### Author Rebuttal · Authors · 2023-08-10
>
> Thank you so much for your reviews and encouragement. The key __contributions__ of our work was to explore the resemblance between the NMDA receptor in the human hippocampus and activation functions in transformers. We are happy to see that the reviewer finds our results convincing.
>
> Here, we would like to address the reviewer’s  __comments__ regarding
> 1) strong statement on the NMDAR-like nonlinearity
> 2) NMDAR impairment experiments
> 3) results on standard ML tasks
> 4) potential evidence of the relationship with the brain
> 5) terminology regarding working memory and episodic memory.
>
> These are important comments, and we believe that addressing them will improve the quality of our work. Please see our responses below:
>
> ---
>
> > __Response 1)__ The NMDAR part is the least convincing, being that standard activation functions work just as well. Don’t think you can have sentences like ‘We find that NMDAR-like nonlinearity is essential for shifting short-term working memory long-term reference memory in transformers’.
>
> As the reviewer noted, the standard activation function ReLU works fine in the transformer model. Our intention was to mention that standard activation functions widely used in deep models such as ReLU, GELU, and Swish can be viewed as a subset of the NMDAR-like activation function (Table 1). We will consider ways to convey this message while toning down our expression, including the following in the abstract:
>
> (Original) We find that NMDAR-like nonlinearity is essential for shifting short-term working memory to long-term reference memory in transformers
> (Revised) We find that NMDAR-like nonlinearity has a beneficial role in shifting short-term working memory to long-term reference memory in transformers
>
> ---
>
> > __Response 2)__ The NMDAR impairment is essentially just making the activation function more linear, which of course will mean that it can’t effectively store long term memories.
>
> Your understanding is correct that making the activation function more linear in feed-forward layers affects the ability of transformers to efficiently store long-term memory. This finding is consistent with neuroscience, which shows that removing the nonlinearity caused by Mg2+ gating affects long-term memory formation [1]. While using linear activation functions may demonstrate limitations in machine learning, this experiment is linked to neuroscientific discoveries. It bridges the gap between understanding the transformer as a memory consolidation model and biological memory processes,  extending beyond traditional machine learning paradigms.
>
> [1] Miyashita et al., (2012). Mg2+ block of Drosophila NMDA receptors is required for long-term memory formation and CREB-dependent gene expression. Neuron, 74(5), 887-898.
>
>
> ---
>
> > __Response 3)__ To effectively make the NMDAR receptor point as a different activation function class, you’d need to test it on more standard ML tasks, rather that bespoke neuroscience tasks.
>
> Thank you for the suggestion. We agree that testing our NMDA function on standard ML tasks would be effective to get NMDAR points as a new function class. We have conducted additional experiments on language modeling and image classification tasks.
> Please refer to our __Global Response 1)__.
>
> ---
>
> > __Response 4)__ You have shown a nice potential neuro-ai link via NMDAR, but it’s currently just a potential relationship. It would need a lot more testing to make it concrete, e.g. showing that α=10 (the best performing model) is the regime that the brain operates in etc.
>
> As suggested by the reviewer, we calculated the corresponding α value in the physiological CA1 hippocampal neurons following real experimental values [1]. The calculated α was ranged between 0.01~0.2.
>
> |Term|Symbol|Typical Value|Units|
> |-|-|-|-|
> | Magnesium Ion Concentration | [ $\text{Mg}^{2+} ]$ |1| mM|
> | Physiological Temperature|T| 37 (310) |°C (K)|
> |Dissociation Constant at $V=0$ (mV) | $K_{\text{Mg}^{2+}}$ | 1 – 20 | mM |
> |Temperature Constant | $\beta$ | 0.062 | mV$^{-1}$ |
>
> These values are typical and can vary depending on the specific biological context, experimental conditions, and the model used. While previous research has shown that increasing the Mg2+ level in the brain with a specific compound such as MgT(magnesium-L-threonate) increases long-term memory formation [2], its effective concentration only increased about 15% from the baseline (expected maximal α in the brain to be ~0.23), possibly due to the physiological ion excretion process in humans.
>
> While the brain's effective α was lower than expected, nonetheless, we are grateful to the reviewer for this suggestion.
>
> [1] Kirson et al., (1999). Early postnatal switch in magnesium sensitivity of NMDA receptors in rat CA1 pyramidal cells. The Journal of Physiology, 521(Pt 1), 99.
> [2] Slutsky et al., (2010). Enhancement of learning and memory by elevating brain magnesium. Neuron, 65(2), 165-177.
>
> ---
>
> > __Response 5)__ Presumably episodic memory is a better name than working memory, being that you’re saying it’s hippocampal related.
> - Terminology: Episodic memory is concerned with the ability to recall specific events and the contexts in which they occurred (What-Where-When), whereas working memory retains information over short durations to perform cognitive tasks. Therefore we used 'working memory' to emphasize short-term retention and processing in the hippocampus, akin to the limited context in transformers.
> - Why Not Episodic Memory?: We avoided this term as our study lacked the time element (When) essential for episodic memory, focusing on What-Where components instead.
> - Future Work: We plan to investigate What-Where-When aspects of episodic memory in future research on transformer models.
>
> We appreciate the reviewer's feedback and hope this response clarifies our terminology choice.

---

> > ### Comment · Reviewer_FYwU · 2023-08-13
> > **Many thanks for your responses.**
> >
> > Many thanks for your responses and I appreciate your calculation of alpha in real neurons. I still think this is a good paper, and I keep my original score.

---

> > > ### Author Response · Authors · 2023-08-18
> > >
> > > We sincerely thank the reviewer for taking the time to read our responses and providing insightful feedback. The rebuttal process has helped us improve our paper, and we are extremely grateful to the reviewer for continued encouragement and support. We will do our best to incorporate the changes discussed in the revised version. Once again, thank you for the thoughtful feedback!

---

### Official Review · Reviewer_xzxf · 2023-07-06

**Soundness:** 3 good
**Presentation:** 3 good
**Contribution:** 2 fair
**Rating:** 6
**Confidence:** 4

**Summary:**

This paper builds on recent work connecting the hippocampus to the transformer architecture by introducing a new hippocampus-inspired activation function for the transformer's feedforward modules. In a toy navigation task, several empirical investigations show that the choice of this activation function has a large effect on the model's ability to store and recall information from longer-term memory, and that models using brain-like activation functions contain neurons with place cell-like firing patterns.

**Strengths:**

* This paper continues an interesting research direction -- exploring the connections between the hippocampus and the transformer architecture. This is exciting work that is relevant to both machine learning and neuroscience.
* The empirical analysis is presented well with many attractive and easy-to-interpret figures, seems quite thorough, and largely supports the paper's conclusions.

**Weaknesses:**

* The paper presents a convincing argument that the nonlinearity in the feedforward module of a transformer is important for forming and recalling certain types of memories, and that the brain-inspired nonlinearity is among the best. However, it provides essentially no understanding (even intuitively) of why this should be the case.

**Questions:**

* Regarding the above weakness: How might the choice of nonlinearity give rise to the effects (reference memory error rate, place cell score, etc) documented here?
* Why is a recurrent embedding used for actions? What happens if a different embedding is used?
* The definition of place cell score is crucial to the paper and should be in the main body, I think. On a related note, I find the definition given in the appendix difficult to understand. In particular, given how the auxiliary graph $\mathcal G$ is constructed, it seems to me that every vertex will be a direct descendant of $node_k$ since $node_k$ has the highest firing rate. How this (a) measures sensitivity of a neuron to a specific location or (b) resembles the peak method used to identify place cells in the neuroscience literature should be explained more carefully. A figure might also be helpful here.
* The discussion on lines 318-324 notes that the results here agree qualitatively with experimental results from neuroscience. It would be nice to have a figure summarizing these findings, and perhaps comparing against this paper's results,

**Limitations:**

Yes.

---

> ### Author Rebuttal · Authors · 2023-08-10
>
> We are glad that the reviewer recognizes our __contributions__ in 1) connecting the hippocampus to the transformer architecture 2) the convincing empirical analysis which supports the transformer’s longer-term memory formation, and 3) unveiling place cell-like firing patterns in a feed-forward layer, 4) with many attractive and easy-to-interpret figures.
>
> Here, we would like to address the reviewer’s  __comments__ regarding:
> 1) Nonlinearity's importance in memory formation.
> 2) Choice of nonlinearity and its effects.
> 3) Recurrent embedding for actions and alternatives.
> 4) Place cell score definition and explanation.
> 5) Summary figure linking findings to neuroscience.
>
> We believe addressing these comments will enhance our work. Please see our responses below:
>
> ---
>
> > __Response 1)__  It provides essentially no understanding (even intuitively) of why this should be the case.
>
> Based on the reviewer’s comment, we find it important to give a more intuitive evidence for why the proposed nonlinearity is effective.
>
> To provide a better understanding of our work, we present both analytical and empirical results indicating that increasing alpha corresponds to increasing sparsity. Please refer to __Global Response 2)__.
>
> ---
>
> > __Response 2)__ How might the choice of nonlinearity give rise to the effects (reference memory error rate, place cell score, etc) documented here?
>
> The high sparsity properties of NMDA function may contribute to the increased score of place cells (__Figure S13 in attached pdf for rebuttal__). In the case of Transformers, enforcing neuron activation sparsity in MLPs has been found to improve the interpretability or selectivity of a higher percentage of neurons [1]. This evidence could explain why the place cell score increases when 𝛼 is increased. Among the various activation functions, NMDA with 𝛼=10 has the highest Gini index. A comparison of the Gini index among activation functions is shown in the figure.
>
> We will include this result in Appendix.
>
> [1] Elhage et al., (2022). Softmax linear units. Transformer Circuits Thread.
>
> ---
>
> > __Response 3)__  Why is a recurrent embedding used for actions? What happens if a different embedding is used?
>
> A recurrent positional embedding for actions was used to capture the temporal dependencies in the agent's actions. The model can encode information about the agent's previous actions and incorporate it into the current position prediction by using recurrent positional embeddings. This can be particularly useful in tasks where the sequence of actions is important, such as navigation. If a different embedding method is used, such as a non-recurrent learnable positional encoding, the model may still be able to learn the task.
>
> In Appendix A.5, we conducted an experiment that may be relevant to the reviewer's question. In this experiment, we disrupted the embedding layers to make them non-recurrent, effectively preventing them from retaining previous action information in embeddings. The results indicate that working memory error and reference memory error increased significantly (see Fig. 3a and Fig. S3a). However, the behavior observed in this experiment is similar to the trend seen when increasing 𝛼 of NMDA$_{\alpha}$ (see Fig. S3b).
>
> These findings imply that, while path-integrated information from recurrent positional embedding is useful for learning the spatial structure of the map, it is not required nor essential to predict the unvisited node.
>
> This finding supports the idea that working memory is crucial for memory consolidation and that disrupting it can cause impairment in reference memory.
>
> ---
>
> > __Response 4)__ The definition of place cell score is crucial to the paper and should be in the main body. … A figure might also be helpful.
>
> Thank you for the in-depth examination of our place cell section. We agree that giving definitions for the place cell score is important. We had tried various writing styles, including having the definition in the main body. In the current version, we tried to go directly to main results. We would be happy to incorporate this feedback and rearrange our content.
>
> As the reviewer suggested, we believe including a schematic figure to explain how an auxiliary graph is constructed would help readability. As shown in __Figure S1 in attached pdf for rebuttal__, not all nodes are direct descendants as we constructed from the 2d grid map. We are grateful for the reviewers' suggestions to improve our work.
>
> ---
>
> > __Response 5)__ It would be nice to have a figure summarizing that the results here agree qualitatively with experimental results from neuroscience.
>
> We will include a summary figure that connects the transformer (this work) and the brain (experimental findings). Please see our __Figure S11 in attached pdf for rebuttal__.

---

> > ### Comment · Reviewer_xzxf · 2023-08-13
> >
> > Thank you for the thorough response. The new figures are very helpful and I hope they will be included in future versions of this paper. In light of this, I'm increasing my score (5 to 6).

---

> > > ### Author Response · Authors · 2023-08-18
> > >
> > > We sincerely thank the reviewer for taking the time to read our responses and providing insightful feedback. The rebuttal process has helped us improve our paper, and we are happy to hear that the newly added information is helpful. We are incredibly grateful to the reviewer for raising the evaluation score, and we will do our best to reflect the corresponding changes in the revised version. Thank you so much!

---

### Official Review · Reviewer_r1mY · 2023-07-07

**Soundness:** 1 poor
**Presentation:** 3 good
**Contribution:** 3 good
**Rating:** 3
**Confidence:** 4

**Summary:**

Many recent papers have shown a relationship between Transformers and biological structures, particularly the hippocampal formation. This work demonstrates that the types of non-linearities provided by NMDAR dynamics (which have known biological importance) can be beneficial in a Transformer architecture for a toy what-where memory/navigation task.

**Strengths:**

This paper deepens the connection between Transformers and biology, which helps both the neuroscience and machine learning communities in understanding these systems. It may also help improve Transformer models, which are already very popular and performant but may be further improved by inspiration from neuroscience.

**Weaknesses:**

The values of $N$ used in Figure 3 suggest that in this case there is a ceiling effect, i.e., once N is sufficiently large, the benefit of the NMDA-like non-linearity becomes negligible. Further, while error bars are included, I could not find the statistical testing to demonstrate differences between the groups. I think it is important to do such tests to measure the size and significance of any effects.

Although the what-where task is clearly neuroscientifically-inspired and has been used in past similar studies, it is not clear whether anything contained here will generalise to other, more ecologically-valid tasks from neuroscience or more naturalistic real-world data from machine learning. The authors should test on more complex tasks/data.

**Questions:**

1. Different activation functions have different computational footprints. How does the computational footprint of the proposed activation function compare to others and is the trade-off worth the performance increase?

2. Line 318. Why is this surprising?

3. While TEM has been popularised and cited widely, and is discussed here, how is it considered 'state-of-the-art' when it is essentially a very fancy Markov chain model? Or do the authors argue that that's what hippocampus is/does?

**Limitations:**

Limitations need more discussion.

---

> ### Author Rebuttal · Authors · 2023-08-10
>
> We appreciate that the reviewer finds our __contributions__ regarding 1) the connection between transformers and hippocampal formation, including the incorporation of NMDAR nonlinear dynamics, and, 2) its potential benefits for both neuroscience and machine learning communities.
>
> Here, we would like to address the reviewer’s  __comments__ regarding
> 1) ceiling effect via increasing N and lack of statistical testing in Figure 3
> 2) generalization to more complex or real-world tasks
> 3) the computational footprint of the proposed activation function
> 4) clarification on why is surprising in line 318
> 5) TEM's characterization and relevance to the hippocampus.
>
> These are great comments, and we believe that addressing them will improve the quality of our work. Please see our responses below:
>
> ---
>
> > __Response 1)__ Values of N used in Fig 3 suggest there is a ceiling effect. If N is sufficiently large, the benefit of the NMDA-like non-linearity becomes negligible? It is important to do statistical testing to demonstrate differences between the groups to measure significances.
>
> Thank you for the feedback. We agree that conducting statistical tests of significance will provide more convincing evidence. Regarding this comment, we ran additional experiments to the level where statistical analysis was possible. We used a nonparametric statistical test, i.e., Mann-Whitney U test, across all groups in Figure 3. This result is included in provided __(Figure S7 in attached pdf for rebuttal)__. We will include this result in our Appendix.
>
> As shown in Figure S7d-f in attached pdf for rebuttal, the overall significance level of NMDA vs others increases (i.e., yellow color means lower $p$-values). We acknowledge we are unable to fully address the reviewer’s concern about the ceiling effect by increasing N. However, based on our current statistical test results, as suggested by the reviewer, NMDA-like non-linearity appears to be effective even in larger N (up to 64) conditions.
>
> ---
>
> > __Response 2)__ It is not clear whether anything contained here will generalize to other. The authors should test on more complex tasks/data.
>
> Thank you for the suggestion. Testing the proposed activation function over more complex tasks and datasets will strengthen our findings. We have conducted experiments on language modeling and image classification tasks. Please refer to __Global Response 1)__.
>
> ---
>
> > __Response 3)__ How does the computational footprint of the proposed activation function compare to others and is the trade-off worth the performance increase?
>
> The computational footprint of the NMDA function is comparable to that of Swish and requires less computation than GELU. Under the JIT compile feature in PyTorch, the computation speeds of Swish, NMDA, and GELU are essentially the same on an actual GPU.
>
> ReLU offers low computational cost and high memory efficiency in large-language model training, but it is not widely used in language models due to its inefficiency in training stability and learning speed. GELU is widely used, despite its higher computational demands, because of its stable training process and rapid reduction in training loss.
>
> While we did not observe the stability issues associated with ReLU in our work, considering these trade-offs into account when evaluating the proposed activation function will be important. We appreciate the reviewer's insight and will consider these factors in future studies.
>
> ---
>
> > __Response 4)__ Line 318. Why is this surprising?
>
> The role of NMDAR in the CA1 region of the hippocampus is known to be vital for long-term memory formation in neuroscience, leading to discoveries such as place cells (i.e., a finding that won a Nobel Prize).
>
> We were surprised to discover parallels between this process and transformers, a leading architecture in deep learning. We observed similarities such as the use of NMDAR-like nonlinearity, the emergence of place cells in feed-forward networks (FFNs), and the effect of linear activation functions on long-term memory.
>
> To clearly convey these points, we have newly added a figure to illustrate these connections in __Figure S11 in attached pdf for rebuttal__; we will include this figure in Appendix.
>
> ---
>
> > __Response 5)__ While TEM has been popularised and cited widely, and is discussed here, how is it considered 'state-of-the-art' when it is essentially a very fancy Markov chain model? Or do the authors argue that that's what hippocampus is/does?
>
> Thank you for your insightful comment regarding the classification of the Tolman-Eichenbaum Machine (TEM) as a 'state-of-the-art' model.
>
> The navigation problem we addressed is inherently non-Markovian. However, the TEM allows us to render the problem Markovian by including a memory M, an integral part of the model that encompasses location-sensory conjunctions.
>
> The TEM is based on the hypothesis that hippocampal cells encode these conjunctions (p = flatten($x^{T} * g$)) and that memories are rapidly stored in weights M through Hebbian learning ($M = M + p^T * p$). This approach not only explains various neural representations in spatial tasks but also extends to non-spatial tasks.
>
> The ability of TEM to represent abstract spatial relationships via sensory input (lateral entorhinal cells) and abstract locations (medial entorhinal cells) uniquely positions it as a model capable of explaining complex neural phenomena. Its ability to store memories via simple Hebbian learning also underscores its novelty.
>
> We referred to TEM as the 'state-of-the-art' model, not only because of its popularity but also because of its comprehensive ability to mimic hippocampal synaptic potentiation, aligning closely with observed neural representations in both spatial and non-spatial tasks.
>
> We hope this clarifies our rationale behind classifying the TEM as 'state-of-the-art' and we welcome any additional comments. (We will also consider giving it a more neutral name.)

---

> > ### Comment · Reviewer_r1mY · 2023-08-18
> > **Response**
> >
> > Responses 4 and 5 re the least convincing. The authors definitely should not refer to TEM as 'state-of-the-art'. I am also uncovinced about the practicality of this contribution to practical probelsm, i.e. the performance-cost tradeoff. Therefore, claims of this nature should also be revised. I am therefore downgrading my score slightly based on the authors responses.

---

> > > ### Author Response · Authors · 2023-08-18
> > >
> > > We regret hearing that our responses were unsatisfying. We would like to ask whether the reviewer meant to change the score from "borderline accept (5)" to "weak reject (4)" given the comment “I therefore downgrade my score slightly”. Instead, the reviewer has reduced the score to "reject (3)," and we sincerely request the reviewer to reconsider our work.
> > >
> > > Given that the response period is still open, we would like to have the opportunity to address the reviewer's feedback better. As the reviewer initially mentioned, our work contributes to connecting Transformers and biology, which helps the neuroscience and machine learning communities understand these systems. We consider this to be an important effort.
> > >
> > > ---
> > >
> > > **Regarding TEM (feedback 5)**, we would like to clarify that we do not consider this model to fully represent the hippocampus. TEM is just one of the models that explain the generalizability of the hippocampus and capture the relational properties of the states, a class of model related to successor representation (SR) [1] (similar to the SMP model [2]).
> > >
> > > To the best of our knowledge, the recent work, TEM-t, is probably the only work that bridges the powerful transformer model with the hippocampus, and our original intention was to highlight this. Thus, we agree that our sentence may have misled the reviewer. As stated in our previous response, we will change the description of TEM to "a recent model that bridges with transformer". As an interdisciplinary team of computer scientists and neuroscientists, we recognize that our description of related work may not have been complete. We'd be happy to accommodate any further suggestions on the literature.
> > >
> > > [1] Dayan, P. Improving generalization for temporal difference learning: the successor representation. Neural Comput. 5, 613–624 (1993).
> > > [2] Uria, B. et al., The spatial memory pipeline: a model of egocentric to allocentric understanding in mammalian brains. Preprint at bioRxiv (2020).
> > >
> > > ---
> > >
> > > **Regarding feedback 4**, "Line 318. Why is this surprising?"
> > >
> > > To our knowledge, no prior studies have linked the transformer's capability for long-term memory in its feed-forward layer to established neuroscience observations. Our study shows that the transformer model aligns with known experimental findings about the role of NMDAR in the hippocampal CA1 memory formation process. To convey this point more effectively, we tried to explain this exciting viewpoint with a schematic figure in the rebuttal process. We kindly ask the reviewer to offer more context for the question.
> > >
> > > 1. **Conceptual Bridge between Neuroscience and Machine Learning**: Transformers are a deep learning architecture, originally designed to process sequential data in tasks like natural language processing. The connection between these computational models and biological processes in the brain is not inherently obvious. Thus, finding evidence that Transformers can be utilized to model memory consolidation and the dynamics of NMDAR in the hippocampus represents an unexpected bridge between two distinct domains.
> > > 2. **Correspondence with Specific Biological Mechanisms**: The fact that the non-linear dynamics of NMDAR and associated parameters (such as Mg$^{2+}$ gating) have specific correspondences in the transformer model (through the activation functions and the modulation of $\alpha$) is a new finding. This correspondence is not merely a superficial similarity but appears to have functional implications for modeling memory formation and place cell representation, which are critical processes in biological neural systems.
> > > 2. **Consistency with Prior Experiments**: Our findings aligned qualitatively with previous NMDAR impairment experiments in neuroscience and surprised us. It strengthens the connection between the computational model and biological reality and suggests potential insights for both fields. In the context of the existing literature, the ability of a computational model to replicate these known biological effects is unforeseen.
> > > 4. **Potential for Practical Applications**: By deepening the understanding of both Transformers and the underlying biological processes, our findings can further lead to improvements in machine learning algorithms inspired by neuroscience and potentially even insights into biological processes informed by computational models.

---

> > > > ### Author Response · Authors · 2023-08-18
> > > >
> > > > **Regarding feedback 3**, "performance-cost tradeoff"
> > > >
> > > > In Global Response 2), we demonstrated that the NMDA function with alpha = 10 has a better validation loss than the GELU activation function. Here, we will describe the practical computational cost. We used the GPT2 (model size: 124M) PyTorch training code from the karpathy/nanoGPT GitHub repository. We should emphasize that we did not scan a wide range of hyperparameters for the NMDA-based model. The optimal NMDA hyperparameters may differ from those of the GELU-based model. Despite no search for hyperparameters, the result shows improved performance. The results in vision tasks are similar, i.e., there was no hyperparameter search.
> > > >
> > > > In addition, we compared the actual time cost of each activation function using a single NVIDIA RTX A6000 GPU. Our experiments showed that the activation functions in PyTorch exhibit similar time costs in practice. Specifically, in our large-scale GPT-2 experiment utilizing ten NVIDIA TITAN V GPUs, the total training time was 107.9 hours for GELU and 107.4 hours for NMDA. Based on these findings, we conclude that the computational costs are nearly identical in practice. Furthermore, the NMDA function with α=10 performs slightly better across various tasks than the other activation functions we investigated in our work.
> > > >
> > > > Below is the code used for cost evaluation.
> > > >
> > > > ```python
> > > > import math
> > > > import torch
> > > >
> > > > class NMDA(torch.nn.Module):
> > > >     def __init__(self, alpha=1.0):
> > > >         super(NMDA, self).__init__()
> > > >         self.a = math.log(alpha)
> > > >
> > > >     def forward(self, x):
> > > >         return x * torch.sigmoid(x - self.a)
> > > >
> > > > nmda = torch.compile(NMDA(10))
> > > > gelu = torch.compile(torch.nn.GELU())
> > > > relu = torch.compile(torch.nn.ReLU(inplace=True))
> > > > lrelu = torch.compile(torch.nn.LeakyReLU())
> > > >
> > > > dtype=torch.float32
> > > >
> > > > x=torch.empty([2**30], device="cuda", dtype=dtype).normal_()
> > > > torch.cuda.synchronize()
> > > > ```
> > > >
> > > > ```
> > > > %%timeit -n 1000
> > > > y=gelu(x)
> > > > torch.cuda.synchronize()
> > > > ```
> > > > 12.6 ms ± 10.2 µs per loop (mean ± std. dev. of 7 runs, 1,000 loops each)
> > > > ```
> > > > %%timeit -n 1000
> > > > y=nmda(x)
> > > > torch.cuda.synchronize()
> > > > ```
> > > > 12.6 ms ± 9.02 µs per loop (mean ± std. dev. of 7 runs, 1,000 loops each)
> > > > ```
> > > > %%timeit -n 1000
> > > > y=relu(x)
> > > > torch.cuda.synchronize()
> > > > ```
> > > > 12.6 ms ± 941 ns per loop (mean ± std. dev. of 7 runs, 1,000 loops each)
> > > > ```
> > > > %%timeit -n 1000
> > > > y=lrelu(x)
> > > > torch.cuda.synchronize()
> > > > ```
> > > > 12.6 ms ± 2.63 µs per loop (mean ± std. dev. of 7 runs, 1,000 loops each)
> > > >
> > > > ---
> > > > Overall, the rebuttal process has helped us improve our paper, and we are grateful to all reviewers for taking the time to give insightful feedback. We cordially ask that the reviewer to reconsider our paper.

---

### Official Review · Reviewer_TzRp · 2023-07-07

**Soundness:** 3 good
**Presentation:** 3 good
**Contribution:** 2 fair
**Rating:** 5
**Confidence:** 2

**Summary:**

This paper investigates the resemblance between the NMDA receptor (NMDAR) in the hippocampus of the human brain and the activation functions used in the transformer architecture (e.g., ReLU, GELU). Then, this paper presents a new activation function that exhibits similarities to NMDAR. It demonstrates that by adjusting the hyperparameter associated with this activation function, the memory capabilities of transformers can be fine-tuned. A 2D grid navigation experiment with transformers is investigated to examine the working memory and the reference memory.

**Strengths:**

1. This paper explored the famous transformer model from a neuroscience perspective, by drawing connections with the hippocampus in the human brain.

2. This paper first investigates the similarity between the NMDAR in the hippocampus and the activation function such as ReLU and GELU in transformer models.

3. The paper in general is well-written and easy to follow.

**Weaknesses:**


1. Although the proposed idea is interesting and neuro-inspired, the technical contribution seems limited (for ML venues). Based on my understanding, Sec 2.2 is a theoretical review of existing work, and the derivation of the NMDAR activation function in Sec 2.3 (w/. A.3 in appendix) is in general straightforward given previous work.

2. The empirical results on the 2D navigation task seem promising, but it may be worthwhile to explore more general tasks that transformer models are typically applied to, e.g., language tasks, to better validate the efficacy of the proposed activation function.

3. In Figure 3 (a), the result shown in the upper right subfigure (test) demonstrates that the proposed method could not predict the unvisited nodes in the novel map. Are there more detailed explanations for the phenomenon?

**Questions:**


1. The technical contributions of this work could be better elucidated.

2. More practical empirical studies e.g., on real-world large-scale datasets or more general and complicated tasks will be more interesting and helpful.

My final score will largely depend on the rebuttal and the discussion with other reviewers. I am willing to increase my score if the concerns are adequately addressed.


**Limitations:**

I did not find potential negative societal impacts in this work. See the “Weaknesses” section for my concerns.

---

> ### Author Rebuttal · Authors · 2023-08-10
>
> Thank you so much for your reviews. The key __contribution__ of our work was to explore the resemblance between the NMDA receptor in the human hippocampus and activation functions in transformers. We appreciate that the reviewer finds our work to be a novel approach introducing a new activation function that reflects the properties of NMDAR and highlights its potential to fine-tune memory capabilities in transformer models.
>
> Here, we would like to address the reviewer’s __comments__ regarding
> 1) technical contribution to ML venues
> 2) tests on more general and practical tasks, and
> 3) more detailed explanations for specific results.
>
> These are important comments, and we believe that addressing them will improve the quality of our work. Please see our responses below:
>
> ---
>
> > __Response 1)__ The technical contribution seems limited for ML venues. The technical contributions of this work could be better elucidated.
>
> The following items demonstrate our technical contributions in terms of improvements over previous work (items 1 & 2) and new experiments (items 3 & 4).
>
> 1. We designed a spatial navigation task that allows the intuitive separation of short-term memory and long-term memory performance.
> 2. We integrated standard activation functions (ReLU, GELU, Swish) into NMDA$_{\alpha, \beta}$ (Table 1) and proposed additional hyperparameter space, $\alpha$, which can potentially be beneficial for ML tasks.
> 3. (Newly added) We conducted additional experimental tasks for standard vision and language models (ViT and GPT2). See details in __Gloabl Response 1)__ and __Figure S12 in attached pdf for rebuttal__.
> 4. (Newly added) We provide additional sparsity analysis for the NMDA$_\alpha$ activation function. This includes the mathematical intuition and empirical results on how increasing $\alpha$ corresponds to an increase in sparse activation in the FFN population. We believe that this analysis will provide a better understanding of our NMDA nonlinearity to ML venues. See details in __Global Response 2)__ and __Figure S13 in attached pdf for rebuttal__.
>
> ---
>
> > __Response 2)__ It may be worthwhile to explore more general tasks that transformer models are typically applied to, e.g., language tasks, to better validate the efficacy of the proposed activation function. More practical empirical studies e.g., on real-world large-scale datasets or more general and complicated tasks will be more interesting and helpful.
> We agree that exploring more general tasks with our proposed activation function would be more interesting and helpful. Therefore, we have conducted additional experiments on language and image classification tasks.
>
> Thank you for the suggestion. Testing the proposed activation function over more complex tasks and datasets will strengthen our findings. We have conducted the following set of additional experiments on language and image classification tasks.
>
> For the result, please refer to  __Gloabl Response 1)__ and __Figure S12 in attached pdf for rebuttal__.
>
> ---
>
> > __Response 3)__ In Figure 3 (a), the result shown in the upper right subfigure (test) demonstrates that the proposed method could not predict the unvisited nodes in the novel map. Are there more detailed explanations for the phenomenon?
>
> This is a good catch. It is due to the transformer model’s design constraint. The problem in our proposed model is caused by the fixed length of the context window (c=64), as depicted in Figure 2b. Any node that has not been visited by the agent in the previous 64 steps is classified as unvisited, which means it is outside the current context window.
>
> The key difference between our model and the transformer model is the recurrent positional embedding, which encodes the previous action sequence information rather than the preceding sequence of observations. Because of the context window size, our model is unable to access the sensory observations of unvisited nodes via the self-attention mechanism.
>
> We will try to explain this limitation better.

---

> ### Comment · Reviewer_TzRp · 2023-08-14
> **Responses to Authors**
>
> I appreciate the responses from the authors and found the newly added figures from the global response helpful. My second question (More practical empirical studies) is addressed and I am willing to increase my score accordingly.

---

> > ### Author Response · Authors · 2023-08-18
> >
> > We sincerely thank the reviewer for taking the time to read our responses and providing insightful feedback. The rebuttal process has helped us improve our paper, and we are happy to hear that the reviewer finds the newly added information helpful. We are extremely grateful to the reviewer for raising the evaluation score, and we will do our best to reflect the corresponding changes in the revised version.

---

### Author Rebuttal · Authors · 2023-08-10

We are grateful to the reviewers for their insightful comments on our study. All reviewers recognized our contribution to investigating NMDAR-like nonlinearity in the transformer's feed-forward network, which will be beneficial for both communities of ML and neuroscience. Reviewers find our work well-written and easy to follow (__Reviewer TzRp__) many attractive and easy-to-interpret figures (__Reviewer xzxf__). Reviewers also think our empirical analysis results support our paper conclusion well (__Reviewer FYwU & xzxf__).

During the rebuttal period, we made additional figures and conducted additional experiments for addressing the reviewers’ questions and concerns. Highlights include experiments on
1) standard machine learning tasks on ViT and GPT2 (Figure S12),
2) sparsity analysis on activation functions (Figure S13),
3) improved illustrations of place cells (Figure S1),
4) statistical significance tests (Figure S7),
5) and clear comparisons between the hippocampus and the transformer model (Figure S11).

These efforts demonstrate our commitment to addressing the reviewers' questions and enhancing the connection between neuroscience and machine learning.  ___Please find our attached PDF regarding our response.___ Thank you.

---
> __Global Response 1)__ standard machine learning tasks on ViT and GPT2

Testing our NMDA activation function over more complex tasks and datasets will strengthen our findings. We have conducted the following set of additional experiments on language and image classification tasks.

__1. Language modeling with GPT2__

We have tested the NMDA$_\alpha$ on the GPT2 model with the Open Web Text dataset. Our analysis indicates a slight performance improvement. Please find this result of loss curves from __Figure S12 in attached pdf for rebuttal__.

__2. Image classification tasks with ViT__

We also tested the NMDA$_\alpha$ on the ViT model (3 trials for each condition) and found an increasing tendency of performance (although statistically non-significant). The table below reports top-1 test accuracies for the CIFAR-100 and the TinyImageNet datasets.
| **Dataset**     | GELU | NMDA$_{\alpha=10}$ | NMDA$_{\alpha=0}$ |
|--------------------|----------|-------------------------------|-----------------------------|
| **CIFAR100** | $69.92\pm0.34$ | $70.27\pm0.47$ | $49.91\pm0.58$ |
| **TinyImNet** | $54.90\pm0.45$ | $55.72\pm0.03$ | $40.36\pm0.52$ |

We would like to mention that although these results are promising, we are not conclusive in stating that the NMDA function outperforms GELU on all real-world tasks. Future work will include testing our NMDA function on different model sizes of GPT and other language models, as well as testing the pre-trained models on downstream NLP tasks such as reading comprehension, question answering, common sense, MMLU, and BIG-bench.

---

> __Global Response 2)__ sparsity analysis on activation functions

To provide a better understanding of our work, we present both analytical and empirical results indicating that increasing alpha corresponds to increasing the sparsity of the feed-forward layer neuronal population, which supports our last paragraph of results section 3.2 (lines 246-248), and third paragraph of the discussion section (lines 309-311).

__Mathematical intuition__:
Given $ \text{NMDA}_{\alpha} = x / (1+\alpha e^{-x}) $, we can rewrite this function as following:

$
\text{NMDA}_\alpha=x \cdot \frac{1}{1+e^{-(x-c)}}, \\
\text{where,} \alpha=\exp{c}.
$

From the above expression, increasing $\alpha$ corresponds to shifting the sigmoid function towards a positive direction by increasing c, which can be interpreted as increasing the threshold of information gating mediated by the sigmoid function. As a result, an increase in $\alpha$ may cause the sparsity of the downstream population activities.

__Empirical result__:
To confirm the above mathematical intuition, we measured the Gini index [1] to determine the sparsity of the feed-forward layer activities. For each input sequence, we calculate the Gini index $ G = \frac{{\sum_{i=1}^{K} \sum_{j=1}^{K} |x_i - x_j|}}{{2K^2 \bar{x}}} $ where $x_i$ is the i-th neuron’s activation value and $ \bar{x} = \sum_{i=0}^K x_i / K$ is the mean of the activation values in the feed-forward layer ($ K=2048 $ is the total number of neurons in a feed-forward layer).

The Gini index ranges from 0 to 1. When only a few neurons have high activation values and others have low values, the Gini index is close to 1. On the other hand, when most neurons have homogeneous activation values, the Gini index is close to 0 (1 = absolute sparsity, 0 = all activations equal).

As shown in __Figure S13 left in attached pdf for rebuttal__, increasing $ \alpha $ in NMDA causes the Gini index to rise. This increase suggests that population activities become more heterogeneous, resulting in a heavy-tailed distribution. Furthermore, the Gini index of NMDA with an alpha value of 10 is greater than that of the GELU activation function (represented by the dashed line). These results imply that $\text{NMDA}_{\alpha}$ may improve long-term memory formation by increasing the sparsity of activations in the feed-forward layer.

Prior studies [2] investigated the generalization performance and sparsity in overparameterized models. Although the mechanism underlying the emergence of sparse representation in large models is not fully understood, it is worth noting that overparameterized models with sparse representations are simpler models than those with dense representations.

The discussion above will be included in the Appendix.

[1] Miller et al., (ICLR 2021). Divisive Feature Normalization Improves Image Recognition Performance in AlexNet.
[2] Li et al., (ICLR 2022). The Lazy Neuron Phenomenon: On Emergence of Activation Sparsity in Transformers.

---

### Decision · Program_Chairs · 2023-09-21

**Decision:**

Accept (poster)

**Comment:**

Recent studies found parallels between transformers and the hippocampus. This study explores the relationship between NMDA receptor (NMDAR) dynamics in the hippocampus and activation functions used in transformer architectures. A new activation function inspired by NMDAR is introduced, impacting the memory capabilities of transformers. A navigation experiment demonstrates that this function affects working and reference memory processes in transformers, and the models that use this function show brain-like spatial representations.

Most reviewer criticism was addressed during the rebuttal period.